# Structure-Aligned Protein Language Model

## Abstract

Protein language models (pLMs) pre-trained on vast protein sequence databases excel at various downstream tasks but often lack the structural knowledge essential for some biological applications. To address this, we introduce a method to enrich pLMs with structural knowledge by leveraging pre-trained protein graph neural networks (pGNNs). First, a latent-level contrastive learning task aligns residue representations from pLMs with those from pGNNs across multiple proteins, injecting inter-protein structural information. Additionally, a physical-level task integrates intra-protein information by training pLMs to predict structure tokens. Together, the proposed dual-task framework effectively incorporates both inter- and intra-protein structural knowledge into pLMs. Given the variability in the quality of protein structures in PDB, we further introduce a residue loss selection module that uses a small model trained on high-quality structures to select reliable yet challenging residue losses for the pLM to learn. Applying our structure alignment method as a simple, lightweight post-training step to the state-of-the-art ESM2 and AMPLIFY yields notable performance gains on tasks where structural signal is directly relevant, including deep mutational scanning (DMS) fitness prediction and a 59% increase in P@L/5 for ESM2 650M contact prediction on CASP16. We further show that these gains scale with model sizes from 8M to 650M, while broader functional-property benchmarks often remain statistically inconclusive or saturated.

## 1 Introduction

Building on recent progress in natural language processing (Brown et al., 2020; Devlin et al., 2019), researchers have focused on pre-training protein language models (pLMs) on vast databases of protein sequences with masked language modeling (Rives et al., 2021; Hayes et al., 2024; Fournier et al., 2024) and next token prediction (Ferruz et al., 2022). These pLMs learn representations that researchers have demonstrated to hold substantial potential across a variety of biological applications, including protein function annotation, enzyme-catalyzed reaction prediction, and protein classification (Hu et al., 2022). Additionally, Rives et al. (2021) observed that structural information emerged in the models' latent representations without supervision. Nonetheless, while the sequence-only nature of pLMs contributes to their widespread adoption, they often struggle with tasks that require detailed structural insights. For instance, the structure-informed ESM-GearNet outperforms ESM2 by 9.7% on the Human Protein-Protein Interaction classification task (Xu et al., 2022; Su et al., 2024).

Yet this structural advantage is contingent on reliable structural inputs at inference. For proteins containing intrinsically disordered regions (IDRs), which constitute a substantial fraction of the human proteome (Lee et al., 2014), structure-conditioned models face a challenge. The structures assigned to disordered regions by folding models are unreliable, and conditioning on them can therefore mislead the model. We demonstrate this on a new downstream task, liquid-liquid phase separation (LLPS) condensate classification. LLPS is a process by which proteins condense into membraneless organelles, often involving IDR-rich proteins. We find that, as disorder increases, structure-conditioned models such as SaProt and the structure-conditioned branch of ESM-GearNet degrade while sequence-only models do not (§4.3). Overall, this observation is aligned with previous findings that structure models cannot distinguish between disordered and dubious proteins, while sequence models can (Fig. 2 in Fournier et al. (2024)). Both issues motivate the need for methods that can leverage structural insights during training while retaining the flexibility of sequence-only models at inference; methods we call structure alignment.

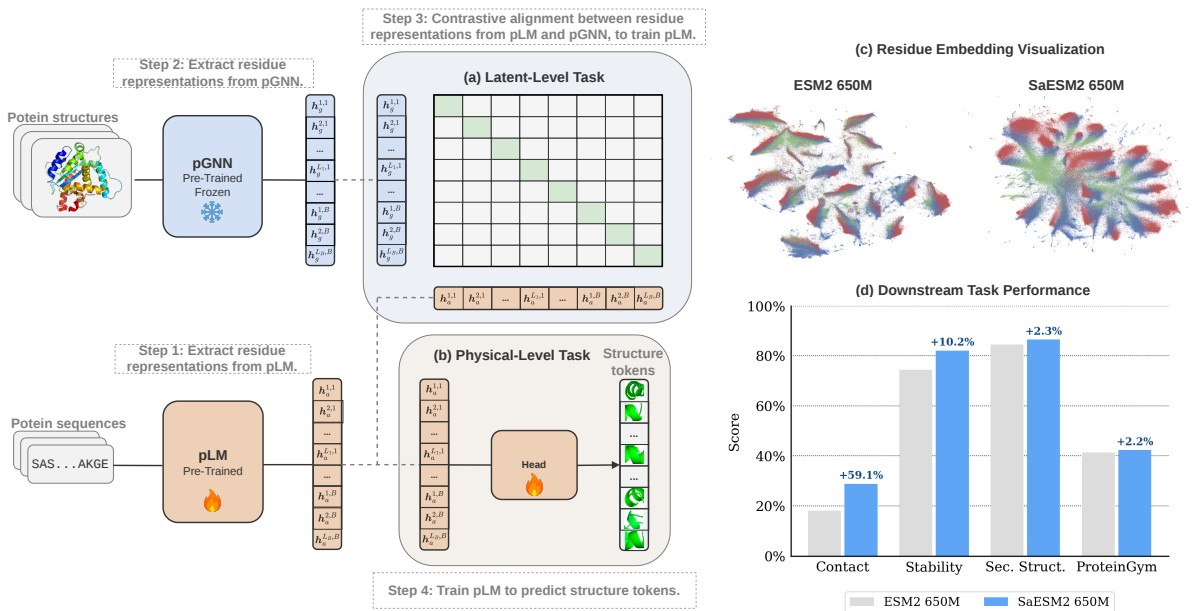

Figure 1: Overview of the *dual-task framework*. (a) Latent-level task: contrastively aligns residue representations from the pLM and pGNN, allowing the pLM to learn inter-protein structural knowledge. (b) Physical-level task: trains the pLM to predict structural tokens, incorporating intra-protein knowledge. (c) Residue embedding visualization: UMAP colored by secondary structure, showing that alignment improves global structure of the feature space. (d) Downstream task performance: structural knowledge improves contact map prediction, thermostability estimation, and fitness landscape modeling.

Given the availability of open-source pre-trained protein graph neural networks (pGNNs) (Zhang et al., 2023; Chen et al., 2023; Jumper et al., 2021), we investigate integrating pGNN-derived structural insights into pLMs. Specifically, we introduce a latent-level contrastive learning task for the structural alignment of pLMs. As illustrated in Figure 1, this task aligns residue hidden representations from the pLM ($\boldsymbol{h}_a$) with those from the pGNN ($\boldsymbol{h}_g$) across a batch of proteins. During this process, the pGNN is frozen while the pLM is optimized to minimize the contrastive learning loss, enriching the pLM with inter-protein structural knowledge, by which we mean residue relationships across proteins in a batch rather than protein–protein interactions. However, pure contrastive alignment may overemphasize differing residue-level patterns across the dataset, neglecting intra-protein structural context, by which we mean structural context within a single protein chain (Zheng & Li, 2024). To address this, we add a physical-level task that trains the pLM to predict structural tokens $\boldsymbol{z}$ (representing physical conformations (van Kempen et al., 2022)) from its residue representations $\boldsymbol{h}_a$. This reinforces the encoding of each residue within its protein, thereby enriching the pLM with intra-protein structural knowledge.

We combine latent and physical tasks, yielding three residue loss types for a batch of proteins with a total length $N$: (i) $N$ sequence-to-structure contrastive losses from the latent-level task, (ii) $N$ structure-to-sequence contrastive losses from the latent-level task, and (iii) $N$ structure token prediction losses from the physical-level task. The *dual-task framework* effectively integrates inter-protein and intra-protein residue-level structural knowledge (§3.1). The masked language modeling loss is additionally incorporated to preserve the sequential knowledge of pLMs.

Given that some protein structure regions in the PDB are ambiguous or inaccurate (Burley et al., 2019), we propose a *residue loss selection* module that prioritizes residue losses aligned with high-quality protein structures across the $3 \times N$ total residue losses (§3.2). First, we use resolution and R-free metrics (Morris et al., 1992) to curate a high-quality reference set and train a small reference model on the set. Next, we compute the *excess loss*, defined as the difference between the residue loss of the current model and that of the reference model (Mindermann et al., 2022). Residue losses with high excess loss are selectively included

in each loss type as they exhibit greater learnable potential. This module filters out inaccurate residues with high reference loss and easy residues with low current loss. By focusing on challenging yet reliable residue losses, the module improves both training effectiveness and efficiency.

We conducted 10 ablations to validate our design choices (Appendix D.5). Our analysis demonstrates that the proposed *dual-task framework* improves performance, with *residue loss selection* providing further gains. The models were evaluated on a comprehensive suite of benchmarks. We assess performance on deep mutational scanning (DMS) fitness prediction using ProteinGym (Notin et al., 2023) and on direct structural validation using a contact prediction task on withheld data from CASP16 (Yuan et al., 2025). We further test generalization on 9 tasks from xTrimoPGLM (Chen et al., 2024) and 9 from SaProt (Su et al., 2024), and evaluate language modeling fidelity using pseudo-perplexity on the high-quality held-out validation set from Fournier et al. (2024). We find that structure alignment is a computationally lightweight post-training step that, depending on the downstream task, either matches or exceeds the performance of the original model.

To summarize, our contributions are as follows:

- We show that existing structure-conditioned models fail when disorder becomes prevalent, while sequence-only models, including structure aligned methods, remain robust.

- We propose a *dual-task framework* that integrates inter-protein and intra-protein residue-level structural knowledge into pLMs, while fully retaining their language modeling capabilities.

- We develop a *residue loss selection* module that prioritizes challenging yet reliable residue losses, thereby improving the learning process and alleviating PDB structure quality issues.

- We conduct extensive experiments that demonstrate the effectiveness of our method across model sizes and model families on some downstream tasks, while identifying a set of benchmarks where performance seems saturated and no method can improve over the original pLMs.

## 2 Preliminaries

In this section, we introduce the preliminaries of protein language models, structure embeddings, and structure tokens used in this study, and provide a more detailed review of related work in Appendix A.

### 2.1 Protein Language Models

Proteins can be represented as amino-acid sequences $\boldsymbol{a} = (a_1, \ldots, a_L)$, where each $a_i$ belongs to one of the 20 common amino acid types. Protein language models are pre-trained on large sequence databases using objectives such as masked language modeling (MLM) (Hayes et al., 2024; Rives et al., 2021) and next-token prediction (Ferruz et al., 2022). We focus on MLM-based pLMs, as proteins are not intrinsically left-to-right, and MLM has been shown to be effective for downstream tasks (Lin et al., 2023b).

We write a pLM with parameters $\boldsymbol{\theta}$ as $\mathrm{pLM}(\cdot; \boldsymbol{\theta})$, with residue representations $\mathrm{pLM}(\boldsymbol{a}; \boldsymbol{\theta}) \in \mathbb{R}^{L \times D_a}$. During pre-training, a subset of positions $\mathcal{M} \subset \{1, \ldots, L\}$ is replaced with a [mask] token to form $\tilde{\boldsymbol{a}}$, and the model is trained to reconstruct the masked amino acids:

$$\mathcal{L}_{\mathrm{mlm}}(\boldsymbol{\theta}, \boldsymbol{\alpha}) = \frac{1}{|\mathcal{M}|} \sum_{i \in \mathcal{M}} \ell_{\mathrm{CE}}\Big( \mathrm{MLP}\left(\mathrm{pLM}(\tilde{\boldsymbol{a}}; \boldsymbol{\theta})_i; \boldsymbol{\alpha}\right), a_i \Big), \tag{1}$$

where $\ell_{\mathrm{CE}}$ is the cross-entropy loss and $\mathrm{MLP}(\cdot; \boldsymbol{\alpha})$ is the prediction head.

### 2.2 Protein Structure Embeddings

Protein language models generate residue-level embeddings from protein sequences. In addition to the sequence perspective, proteins exist as 3D structures, and this physical nature largely determines their biological functions. Recent studies have also investigated deriving residue-level embeddings directly from protein 3D structures. One approach is to use the residue-level hidden representations generated by AlphaFold2 (Jumper

et al., 2021), although their effectiveness for downstream tasks has since been questioned (Hu et al., 2022). GearNet (Zhang et al., 2023) addresses this limitation by pre-training a protein graph model encoder using multiview contrastive learning. Similarly, STEPS (Chen et al., 2023) improves protein structural representations by introducing multiple self-prediction tasks during graph model pre-training.

Given a protein graph $\boldsymbol{g}$, where each residue is a node and edges are defined based on both sequential and structural distances, a pre-trained protein GNN model outputs residue-level embeddings $\text{pGNN}(\boldsymbol{g}) \in \mathbb{R}^{L \times D_g}$, where $L$ is the number of residues, and $D_g$ is the embedding dimension of the graph-based residue representation.

### 2.3 Protein Structure Tokens

Inspired by the success of token-based protein language models, recent studies have explored the idea of tokenizing protein structures, representing a protein's 3D conformation as a series of discrete structure tokens. A protein structure with $L$ residues can be expressed as $\boldsymbol{z} = (z_1, z_2, \ldots, z_L)$, where $z_i$ denotes the structure token for the $i$-th residue.

Foldseek (van Kempen et al., 2022) introduces an efficient method for tokenizing protein structures, where each residue $i$ is described by its geometric conformation relative to its spatially closest residue $j$. While this approach has significantly accelerated homology detection, it incurs substantial information loss, thereby limiting its applicability to tasks requiring detailed structural reconstruction. To address this limitation, ProToken (Lin et al., 2023a) employs a symmetric encoder-decoder architecture that enables high-fidelity reconstruction of protein structures from tokens. Despite this advancement, these tokens have shown limited effectiveness in broader downstream applications (Zhang et al., 2024a).

Recently, Hayes et al. (2024) developed an effective vector quantization variational autoencoder (VQ-VAE) tokenizer and integrated structure and sequence into a multi-modal protein language model called ESM3. This approach effectively combines both modalities, improving the model's versatility. While we could not evaluate ESM3 ourselves due to licensing restrictions, we were able to retrieve its reported performance on the ProteinGym benchmark. AIDO (Zhang et al., 2024a) further enhances structure tokenization by introducing a novel VQ-VAE with an equivariant encoder and an invariant decoder, ensuring a more robust representation of protein structures.

## 3 Method

### 3.1 Dual-Task Framework

**Latent-Level Task** To incorporate structural insights from pre-trained pGNNs, we propose a latent-level contrastive learning task for the structure alignment of pLMs. Assuming a batch contains $B$ proteins, with a total of $N = \sum_{b=1}^{B} L_b$ residues, we perform contrastive learning across all residues. We denote the pLM hidden representation of the $i$-th residue from the $b_1$-th protein sequence $\boldsymbol{a}_{b_1}$ as $\text{pLM}(\boldsymbol{a}_{b_1}; \boldsymbol{\theta})_i$, and the pGNN hidden representation of the $j$-th residue from the $b_2$-th protein structure $\boldsymbol{g}_{b_2}$ as $\text{pGNN}(\boldsymbol{g}_{b_2})_j$. Note that the parametrization of the pGNN is omitted for brevity, as the pGNN is frozen during training while only the pLM parameters $\boldsymbol{\theta}$ are optimized.

To align these embeddings, we introduce two linear layers, $\boldsymbol{W}_a \in \mathbb{R}^{D_a \times D}$ and $\boldsymbol{W}_g \in \mathbb{R}^{D_g \times D}$, which project both embeddings into the same dimension $D$, together with a learnable scalar $s$. We use $\boldsymbol{W}$ as shorthand for the learned projection parameters and scalar temperature. The similarity score between residues is computed as:

$$\delta_{i,j}(b_1, b_2) = s\big(\text{pLM}(\boldsymbol{a}_{b_1}; \boldsymbol{\theta})_i \boldsymbol{W}_a\big)^{\top}\big(\text{pGNN}(\boldsymbol{g}_{b_2})_j \boldsymbol{W}_g\big), \tag{2}$$

where $s$ follows the approach in CLIP (Radford et al., 2021). The contrastive step is similar in spirit to that of Robinson et al. (2023), with three notable exceptions: (i) we keep the pGNN frozen, since our objective is to align the pLM; (ii) we retain the language modeling head, which they discard but our final loss Equation 8 relies on; and (iii) we keep the physical-level task introduced below, which they replace with another, higher-level, inter-protein contrastive loss. Their objective also differs from ours, as they focus on improving sequence models without preserving the original language-modeling interface.

The sequence-to-structure residue contrastive loss for the $i$-th residue in the $b_1$-th protein is:

$$\mathcal{L}_{\text{a2g}}(\boldsymbol{\theta}, \boldsymbol{W}, i, b_1) = -\log \frac{e^{\delta_{i,i}(b_1, b_1)}}{\sum_{b_2=1}^{B} \sum_{j=1}^{L_{b_2}} e^{\delta_{i,j}(b_1, b_2)}}. \tag{3}$$

The sequence-to-structure contrastive loss for the batch is then:

$$\mathcal{L}_{\text{a2g}}(\boldsymbol{\theta}, \boldsymbol{W}) = \frac{1}{N} \sum_{b_1=1}^{B} \sum_{i=1}^{L_{b_1}} \mathcal{L}_{\text{a2g}}(\boldsymbol{\theta}, \boldsymbol{W}, i, b_1). \tag{4}$$

A similar residue loss, $\mathcal{L}_{\text{g2a}}(\boldsymbol{\theta}, \boldsymbol{W}, j, b_2)$, can be defined for structure-to-sequence contrast, leading to the overall structure-to-sequence loss $\mathcal{L}_{\text{g2a}}(\boldsymbol{\theta}, \boldsymbol{W})$. The final latent-level loss is then given by:

$$\mathcal{L}_{\text{latent}}(\boldsymbol{\theta}, \boldsymbol{W}) = \frac{1}{2}\big(\mathcal{L}_{\text{a2g}}(\boldsymbol{\theta}, \boldsymbol{W}) + \mathcal{L}_{\text{g2a}}(\boldsymbol{\theta}, \boldsymbol{W})\big), \tag{5}$$

which enhances the pLM by integrating inter-protein residue-level structural knowledge.

In our experiments, we primarily use GearNet (Zhang et al., 2023) as the pGNN, pre-trained on the AlphaFold2 database (Varadi et al., 2022). We also evaluated the Evoformer within AlphaFold2 (Jumper et al., 2021) but found GearNet embeddings to be more effective for our purpose.

**Physical-Level Task**  The contrastive objective uses every other residue in the batch as a negative, so its signal is dominated by residue discrimination across proteins. It does not directly require a residue embedding to encode the local geometry of its own chain. To add this within-protein signal, we introduce a physical-level task to reinforce the encoding of residue structure relative to its own protein.

This task trains the pLM to use the residue hidden representation to predict its structural token $\boldsymbol{z}$, which represents the residue's physical conformation (van Kempen et al., 2022). The structure token prediction loss for the $i$-th residue in the $b_1$-th protein is defined as:

$$\mathcal{L}_{\text{physical}}(\boldsymbol{\theta}, \boldsymbol{\beta}, i, b_1) = \ell_{\text{CE}}\Big(\text{MLP}\big(\text{pLM}(\boldsymbol{a}_{b_1}; \boldsymbol{\theta})_i; \boldsymbol{\beta}\big), z_{i,b_1}\Big) \tag{6}$$

where $\ell_{\text{CE}}$ denotes the cross-entropy loss, MLP represents a multi-layer perceptron, and $\boldsymbol{\beta}$ are the parameters of the MLP. The overall physical-level loss is given by:

$$\mathcal{L}_{\text{physical}}(\boldsymbol{\theta}, \boldsymbol{\beta}) = \frac{1}{N} \sum_{b_1=1}^{B} \sum_{i=1}^{L_{b_1}} \mathcal{L}_{\text{physical}}(\boldsymbol{\theta}, \boldsymbol{\beta}, i, b_1), \tag{7}$$

infusing the pLM with intra-protein residue-level structural knowledge.

In our experiments, we use Foldseek (van Kempen et al., 2022) tokens and also evaluated Protokens (Lin et al., 2023a) and AIDO (Zhang et al., 2024a).

**Overall Loss**  In addition to the dual-task losses, we incorporate the original MLM loss to preserve the sequential knowledge of pLMs, resulting in the final loss function:

$$\mathcal{L}(\boldsymbol{\theta}, \boldsymbol{\alpha}, \boldsymbol{W}, \boldsymbol{\beta}) = \mathcal{L}_{\text{mlm}}(\boldsymbol{\theta}, \boldsymbol{\alpha}) + \gamma_l \mathcal{L}_{\text{latent}}(\boldsymbol{\theta}, \boldsymbol{W}) + \gamma_p \mathcal{L}_{\text{physical}}(\boldsymbol{\theta}, \boldsymbol{\beta}), \tag{8}$$

We set $\gamma_l = \gamma_p = 0.5$, giving equal weight to the latent-level and physical-level tasks while keeping the MLM term at its native scale.

Among similarly structure-aligned or contrastively fine-tuned sequence-only pLMs, our method keeps the MLM head and can therefore be evaluated directly on language-modeling downstream tasks, such as Deep Mutation Scanning, which are crucial in drug discovery pipelines. Also, contrary to existing models such as SaProt (Su et al., 2024), there is no need for structure as input to enrich residue embeddings. This structure-agnostic capability is essential, given that proteins with intrinsically disordered regions, which lack a fixed tertiary structure, constitute a significant portion of the proteome. Independence from structural input ensures the model's applicability to any protein, including those with uncharacterized structures or those for which in silico folding predictions may be unreliable.

### 3.2 Residue Loss Selection

To address the challenge posed by ambiguous or inaccurate protein structures in the PDB (Burley et al., 2019), we propose a *residue loss selection* module. This module ensures both effectiveness and efficiency by prioritizing residue losses that align with high-quality protein structures.

**Reference Set**   We begin by curating a high-quality reference set using resolution and R-free metrics (Morris et al., 1992). Structures with a resolution below 2.0Å and an R-free value below 0.20 are selected as a clean reference set. We then train a smaller language model on the reference set with the same loss in Equation 8 and denote the optimized reference model parameters as $\boldsymbol{\theta}^{\mathrm{r}}$, $\boldsymbol{\alpha}^{\mathrm{r}}$, $\boldsymbol{W}^{\mathrm{r}}$, $\boldsymbol{\beta}^{\mathrm{r}}$. For each reference model, we choose to downsize each model in a family once, aligning ESM2-650M using a reference post-trained from ESM2-150M, a 35M for the 150M and so on. The resulting reference model is used to assess the residue loss of the alignment corpus.

**Excess Loss**   For each residue loss discussed in §3.1, we compute the *excess loss*, defined as the difference between the residue loss of the current model and that of the reference model:

$$\Delta\mathcal{L}_{\mathrm{a2g}}(i, b_1) = \mathcal{L}_{\mathrm{a2g}}\big(\boldsymbol{\theta}, \boldsymbol{W}, i, b_1\big) - \mathcal{L}_{\mathrm{a2g}}\big(\boldsymbol{\theta}^{\mathrm{r}}, \boldsymbol{W}^{\mathrm{r}}, i, b_1\big),$$
$$\Delta\mathcal{L}_{\mathrm{g2a}}(j, b_2) = \mathcal{L}_{\mathrm{g2a}}\big(\boldsymbol{\theta}, \boldsymbol{W}, j, b_2\big) - \mathcal{L}_{\mathrm{g2a}}\big(\boldsymbol{\theta}^{\mathrm{r}}, \boldsymbol{W}^{\mathrm{r}}, j, b_2\big), \tag{9}$$
$$\Delta\mathcal{L}_{\mathrm{physical}}(i, b_1) = \mathcal{L}_{\mathrm{physical}}\big(\boldsymbol{\theta}, \boldsymbol{\beta}, i, b_1\big) - \mathcal{L}_{\mathrm{physical}}\big(\boldsymbol{\theta}^{\mathrm{r}}, \boldsymbol{\beta}^{\mathrm{r}}, i, b_1\big).$$

where $\Delta\mathcal{L}_{\mathrm{a2g}}(i, b_1)$, $\Delta\mathcal{L}_{\mathrm{g2a}}(j, b_2)$, and $\Delta\mathcal{L}_{\mathrm{physical}}(i, b_1)$ represent the residue excess loss for sequence-to-structure, structure-to-sequence, and physical tasks, respectively.

**Loss Selection**   Residue losses with high excess loss are prioritized for inclusion in the training as they exhibit greater learnable potential. This effectively filters out inaccurate residues, which typically have high reference model loss, and excludes easy residues with low current model loss. We introduce a selection ratio $\rho$, selecting $N\rho$ residue losses for each type of loss. Taking $\mathcal{L}_{\mathrm{a2g}}$ as an example, we rewrite Equation 4 using the "active" index set $\mathcal{I}_{\rho}$ of the top $\rho$ of all $\Delta\mathcal{L}_{\mathrm{a2g}}(i, b_1)$ values

$$\mathcal{L}_{\mathrm{a2g}}(\boldsymbol{\theta}, \boldsymbol{W}) = \frac{1}{|\mathcal{I}_{\rho}|} \sum_{(i, b_1) \in \mathcal{I}_{\rho}} \mathcal{L}_{\mathrm{a2g}}(\boldsymbol{\theta}, \boldsymbol{W}, i, b_1). \tag{10}$$

This selection process is applied similarly for the other two types of losses. By focusing on challenging yet reliable residue losses, the *residue loss selection* module improves overall training effectiveness and efficiency.

## 4 Experiments

### 4.1 Structure Alignment Details

We aligned ESM2 and AMPLIFY using $129,732$ proteins from OpenFold (Ahdritz et al., 2023) present in the PDB database, of which $116,713$ are for training and $13,019$ for validation. We systematically verified that our sequences were deposited in the PDB no later than December 2021. As a consequence, our training set is fully deduplicated against CASP16, making it a gold-standard test set for downstream evaluation of our models.

The training protocol is adapted from the AMPLIFY stage-2 configuration (Fournier et al., 2024) with several modifications. We extend the pre-training with 20 epochs on our alignment dataset, with the learning rate linearly warming up from 0 to the peak rate over the first two epochs, followed by a cosine decay schedule for the subsequent 18 epochs. The peak rate for the language model is set at $1 \times 10^{-4}$, as per the AMPLIFY standard, while other modules, such as the structural linear classifier and the contrastive learning module, are set at $1 \times 10^{-3}$. The selection ratio $\rho$ is set to 0.8.

We employ the Zero Redundancy Optimizer (ZeRO) with DeepSpeed and use 8 H100 GPUs. The effective batch size is $4,096$ samples at a sequence length of $2,048$, with longer proteins being randomly truncated. Our post-training alignment method is particularly compute efficient, taking under 6 hours for the largest ESM2 model considered, and under 1 hour for the smallest model.

## 4.2 Baseline Models

We evaluate the following sequence-only baseline pLMs: (1) **ESM2**: the standard ESM2 650M model (Lin et al., 2022); (2) **AMPLIFY**: the standard AMPLIFY 350M model (Fournier et al., 2024); (3) **ESM2-S**: a variant of ESM2 fine-tuned for fold classification (Zhang et al., 2024b); (4) **ISM**: a variant of ESM2 optimized for structure token prediction (Ouyang-Zhang et al., 2024); (5) **S-PLM**: a different contrastive post-training method applied to ESM2 (Wang et al., 2025), with approximately 100M additional parameters compared to ESM2 and the other ESM2-based baselines. We denote our structure-aligned ESM2 and AMPLIFY models as **SaESM2** and **SaAMPLIFY**.

## 4.3 Intrinsically Disordered Proteins: Condensate Membership

**Task**   We use the DrLLPS database (Ning et al., 2020), which catalogs proteins annotated for membership in specific biomolecular condensates, as a benchmark for evaluating model behaviour across the disorder spectrum. We frame the problem as multilabel condensate classification: given a protein's embedding, predict which condensate type(s) it belongs to. We retain the 13 condensate types with at least 50 annotated proteins, covering 8,969 sequences in total. We evaluate on a held-out test split whose proteins all have MobiDB disorder annotations (Piovesan et al., 2025); these annotations define the per-protein disorder scores used in the analysis. We deduplicate the test set at 50% sequence identity and 80% coverage. We use IUPred2A (Mészáros et al., 2018) only to verify that the train and test splits have similar disorder-level distributions. An XGBoost classifier is trained on each model's embeddings, and micro-F1 is reported on the test split as a function of MobiDB disorder score, estimated with a kernel smoother.

**Structure-conditioned models fail under disorder**   The benchmarks that dominate pLM evaluation are skewed towards well-folded proteins, precisely the regime where structure-conditioned models have an inherent advantage. As shown in Figure 2 (top), this advantage evaporates as disorder increases: SaProt with Foldseek tokens derived from ESMFold-predicted structures trends downward at higher disorder levels, and the structure-conditioned branch of ESM-GearNet on ESMFold-predicted structures straight out underperforms. Comparatively, sequence-only counterparts (ESM2, AMPLIFY) maintain either high, stable performance or even improve as disorder increases. SaProt in wildcard mode, without structure tokens, underperforms on well-folded proteins but holds up better at higher disorder levels.

**Structure alignment preserves the sequence-only advantage**   As shown in Figure 2 (bottom), none of the structure-aligned models exhibit the disorder-induced degradation observed in structure-conditioned models, confirming that incorporating structural knowledge at training time without requiring structural input at inference preserves the sequence-only robustness on IDPs.

## 4.4 Supervised Downstream Task Performance

To evaluate the effectiveness of our structure alignment, we benchmark our models against their unaligned counterparts on a comprehensive suite of supervised downstream tasks. For these tasks, the pLM is fine-tuned, either with a head and frozen backbone or with full-model fine-tuning, for each specific objective. We group these into structural property prediction, supervised mutation effect prediction, and broader functional property prediction.

Whenever possible, we report either 95% confidence intervals computed via bootstrapping for downstream property prediction tasks, or the full distributions across the 217 sub-benchmark assays of ProteinGym (see §4.5). These bootstrap intervals quantify test-set sampling uncertainty; post-training seed stability is reported separately in Appendix D.4.

### 4.4.1 Structural Property Prediction

**Tasks**   To test the hypothesis that structure-aligned models capture more nuanced insights of protein structures, we evaluate on the following structure prediction tasks from xTrimoPGLM (Chen et al., 2024): (1) **Contact**: two residues from the same chain are considered in contact if their $C_\alpha$ atoms lie within 8Å (Rao

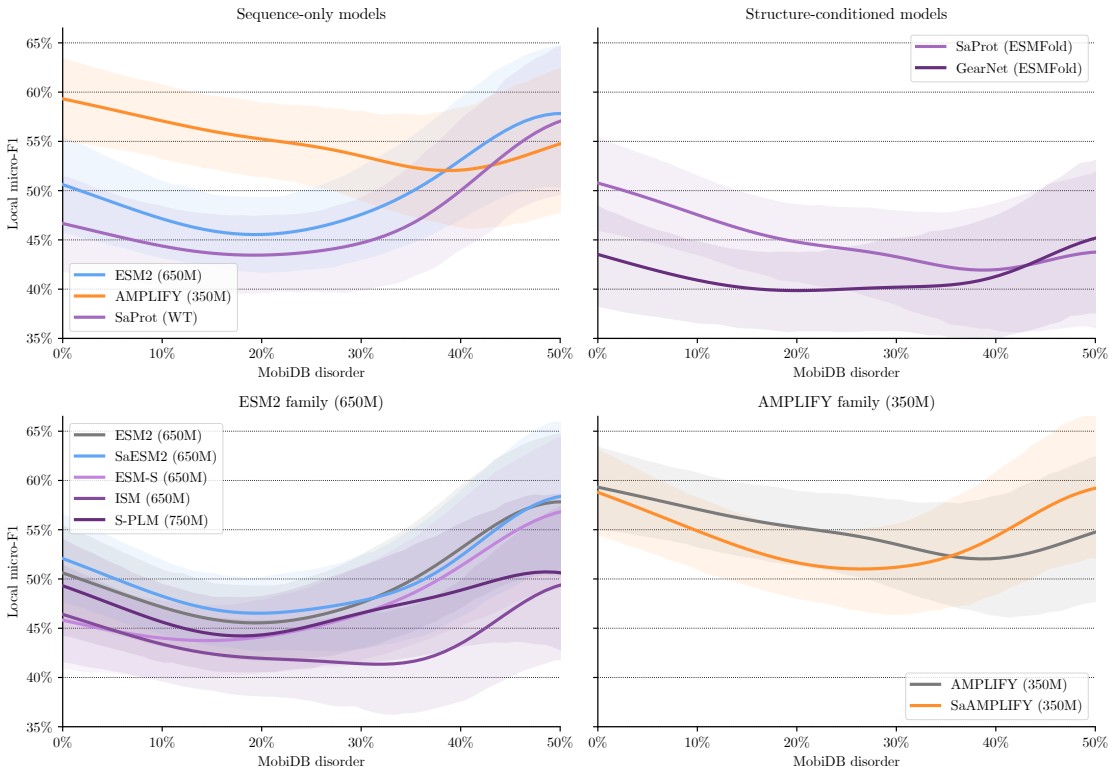

Figure 2: Condensate classification micro-F1 (XGBoost) as a function of MobiDB disorder score, estimated with a kernel smoother. **(top)** Sequence-only models remain optimal or improve across the disorder spectrum, while structure-conditioned models derived from ESMFold-predicted structures, including SaProt and ESM-GearNet without final ESM2 embedding concatenation, underperform at high disorder levels with performance degradation for SaProt. **(bottom)** Structure-aligned models preserve the sequence-only advantage. Shaded bands are 95% bootstrap confidence intervals.

et al., 2019). We evaluate this task using Top L/5 precision, as Rives et al. (2021), considering intra-chain residue pairs with a sequence separation greater than 6 and a sequence length cutoff of 512. In order to compare existing models without data leakage, we select the subset of CASP16 proteins that have already been deposited in PDB and contain at least one long-range intra-chain contact. This yields 51 chains from 13 complexes. When a PDB entry contains multiple chains or biological assemblies, we keep each chain-level target, which introduces some redundancy within the CASP16 split; the reported confidence intervals are computed by bootstrapping over the evaluated chain-level targets. We also report the original xTrimoPGLM test split (created from the trRosetta dataset), and (2) **Fold Classification (Fold)**: classify each protein sequence into one of $1,195$ fold classes (Hou et al., 2018), with accuracy as the evaluation metric. (3) **Secondary Structure (SS)**: assign each residue to one of three secondary structure types (Rao et al., 2019), using accuracy as the evaluation metric.

To assess the quality of the learned representations, *we freeze the backbone model and train a linear head* for 20 epochs using a batch size of 128. We use a learning rate of $1 \times 10^{-3}$, with betas set to $(0.9, 0.95)$ and a weight decay of 0.01 (Fournier et al., 2024). The linear head has a hidden size of 128, following the methodology of xTrimoPGLM. The linear head operates on residue embeddings for the token-level task (SS), on the mean-pooled residue embedding for the sequence-level task (Fold), and on pairwise residue embedding for the Contact task. We further visualize residue embeddings with secondary structure labels to assess the effectiveness of structural alignment in Appendix C.

**Analysis**   As shown in Table 1, SaESM2 and SaAMPLIFY outperform their respective base models on all structure prediction tasks as well as existing alignment baselines on two out of three tasks, improving Contact P@L/5 on CASP16 by 59% for ESM2 and 9% for AMPLIFY. This is a direct validation of the way inter- and intra-protein structural knowledge is infused by our method. Because ESM2-S was directly trained on fold classification with an unfrozen backbone, it outperforms our alignment method on the corresponding task.

In Appendix D.1, we show that these conclusions still hold across model size and families for secondary structure prediction and, to a lesser extent, fold classification, providing evidence that the method scales.

Table 1: Results on supervised downstream tasks. We report the primary metric for each task. Values are formatted as Metric [95% Confidence Interval]. GB1 Fitness and Stability are supervised xTrimoPGLM tasks; zero-shot ProteinGym DMS results over 217 assays are reported separately in §4.5. The best-performing model within each family (ESM2-based and AMPLIFY-based) is in **bold**. Models whose mean falls within the 95% confidence interval of the best-performing model in their family are in *italic*.

| Model | Contact (P@L/5 ↑) | | Fold | SS | GB1 Fitness | Stability |
| | trRosetta | CASP16 | Acc (↑) | Acc (↑) | Sp. (↑) | Sp. (↑) |
|---|---|---|---|---|---|---|
| ESM2 | 0.390 [0.380, 0.400] | 0.181 [0.146, 0.224] | 0.677 [0.662, 0.692] | 0.845 [0.843, 0.847] | 0.945 [0.937, 0.951] | 0.744 [0.736, 0.752] |
| ESM2-S | 0.387 [0.377, 0.398] | 0.182 [0.148, 0.222] | **0.764** [0.750, 0.778] | 0.811 [0.809, 0.813] | **0.961** [0.955, 0.965] | 0.765 [0.756, 0.773] |
| ISM | 0.426 [0.417, 0.436] | 0.220 [0.181, 0.262] | 0.598 [0.580, 0.614] | 0.840 [0.838, 0.842] | *0.957* [0.951, 0.962] | 0.558 [0.545, 0.571] |
| S-PLM | 0.403 [0.394, 0.413] | 0.229 [0.190, 0.273] | 0.662 [0.646, 0.677] | 0.821 [0.819, 0.823] | 0.947 [0.940, 0.952] | 0.661 [0.651, 0.672] |
| SaESM2 | **0.461** [0.450, 0.471] | **0.288** [0.250, 0.327] | 0.681 [0.665, 0.696] | **0.865** [0.863, 0.866] | *0.957* [0.951, 0.962] | **0.820** [0.813, 0.827] |
| AMPLIFY | 0.253 [0.245, 0.262] | *0.155* [0.120, 0.192] | 0.487 [0.470, 0.502] | 0.811 [0.809, 0.813] | *0.947* [0.941, 0.953] | 0.713 [0.704, 0.722] |
| SaAMPLIFY | **0.320** [0.311, 0.328] | **0.169** [0.144, 0.195] | **0.576** [0.557, 0.593] | **0.849** [0.847, 0.850] | **0.948** [0.941, 0.953] | **0.747** [0.739, 0.756] |

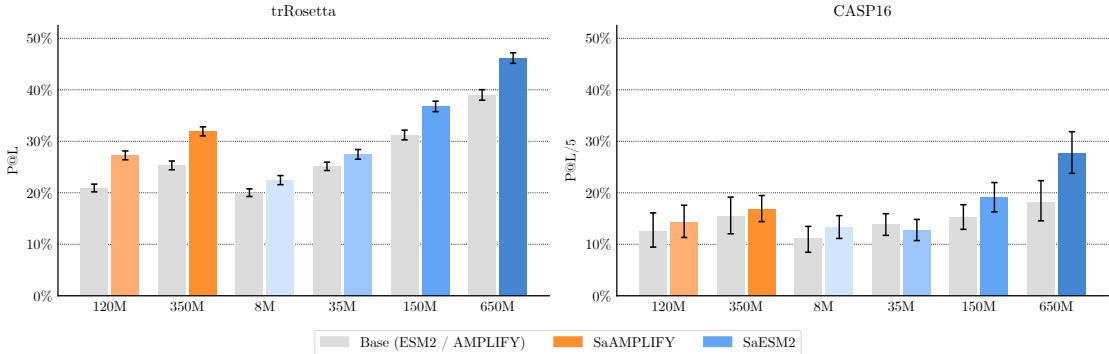

Figure 3: Long-range intra-chain contact prediction precision on the CASP16 test set and the trRosetta test split from the xTrimoPGLM evaluation pipeline. Structure-aligned models significantly outperform their baseline on both test sets, starting at the 150M parameter model size. Error bars show 95% bootstrap confidence intervals over test-set targets; confidence intervals on CASP16 are larger because this split contains 51 evaluated chains from 13 complexes.

### 4.4.2 Functional Property Prediction

We evaluate SaESM2 and SaAMPLIFY on a broad suite of downstream property prediction tasks (Xu et al., 2022; Dallago et al., 2021), which rely on structural information to some extent but are not direct structure prediction tasks. These include predictions of thermostability, metal ion binding, protein localization (DeepLoc), enzyme commission numbers (EC), gene ontology annotations (GO), and protein–protein interactions (HumanPPI) for tasks and evaluation pipeline from (Su et al., 2024).

In addition to downstream structural evaluations (§4.4.1) and supervised mutation effect prediction (§4.4.3) from xTrimoPGLM, we further evaluate on 3 other properties: enzyme catalytic efficiency, peptide-MHC/TCR binding affinity, and peptide-HLA-MHC affinity. In all cases, we follow the data splits and training protocols from the respective papers. The last two tasks concern immunology and specifically interactions between

peptides (short proteins), Major Histcompatibility Complexes (MHC), also called Human Leukocyte Antigen in humans and T-Cell Receptor, driving adaptive immunity.

As detailed in §D.1 and §D.2, these functional-property benchmarks show a different pattern from the structure-proximal tasks. We do not observe statistically reliable gains from structure alignment on most of them, but we also do not observe a statistically reliable degradation where uncertainty estimates are available. Notably, the confidence intervals of the three ESM2-size-preserving structure-aligned variants (SaESM2, ESM2-S, and ISM) fully overlap with those of their baseline across all 9 SaProt tasks, except GO-CC. S-PLM is reported in the same tables but excluded from this count because it adds approximately 100M parameters. Out of these 27 points of comparison, the structure-aligned models outperform their baselines only 14 times, barely over half.

This contrast suggests that the useful signal differs across benchmark families. Reinforcing structural information during post-training yields consistent gains on tasks close to geometry or protein fitness landscapes, including contact prediction, secondary structure prediction, supervised Stability, and zero-shot DMS. DMS and Stability are not direct structure prediction tasks, but their labels can be strongly constrained by whether mutations preserve a viable fold, interaction surface, solubility, or other biophysical properties. By contrast, broader function benchmarks appear more heterogeneous: some show little scaling with model size, whereas others improve with stronger base sequence models but not with the structure-alignment variants evaluated here. This pattern is compatible with pLMs carrying partly separable functional and structural signals, rather than a single representation in which reinforcing structural information necessarily improves every functional readout. Figure 4 provides a compact descriptive summary of these average scores, with task-level details in §D.1 and §D.2.

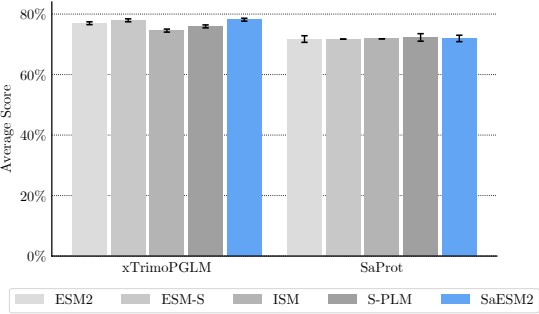

Figure 4: Average performance across xTrimoPGLM and SaProt tasks for 650M-scale models. The modest average improvement hide a large variability in downstream task performance.

### 4.4.3 Supervised Mutation Effect Prediction

**Tasks**  We evaluate our models on protein mutation effect prediction. Specifically, we consider two supervised tasks adopted in xTrimoPGLM: (1) **Fitness (GB1)**: predicting the binding fitness of GB1 following mutations at four specific positions; (2) **Stability**: predicting relative protease resistance as a proxy measurement for stability. For this task, evaluation is performed on one-mutation neighborhoods of the most promising proteins (Rao et al., 2019). We report performance using the Spearman correlation coefficient, while the setup is the same as in §4.4.1, except that we also fine-tune the backbone with a learning rate of $1 \times 10^{-4}$.

**Analysis**  As shown in Table 1, SaESM2 demonstrates a clear advantage over other models in supervised mutation effect prediction in Stability prediction [1] and a statistically competitive performance for Fitness prediction (GB1).

---

[1]Note that standard deviation of the downstream evaluation pipeline for Stability prediction is very high (see Table 5 in Appendix)

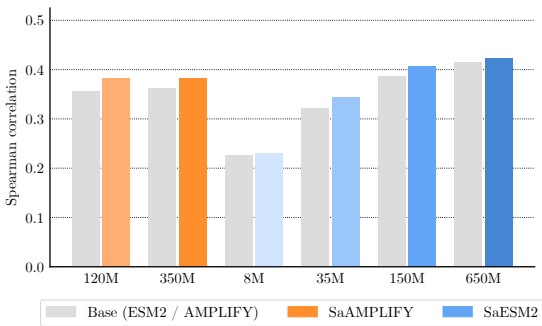

Figure 5: DMS average Spearman correlation across 217 ProteinGym substitution assays for different families of models. Structure-aligned models, identified by the prefix Sa-*, consistently outperform their baseline models. Full assay-level distributions and the size-performance comparison with ESM-C and ESM3 reference points are shown separately in Appendix D.3.

Additional results across model sizes and downstream tasks can be found in Appendix D.1, providing further insights into how the method scales. In the same section, we also report our statistically inconclusive results on Fluorescence prediction under protein mutations.

### 4.5  Zero-Shot Deep Mutational Scanning

**Tasks**  We evaluate all our models on zero-shot deep mutational scanning (DMS), which is related to the supervised GB1 fitness and Stability tasks in §4.4.3 but follows a distinct evaluation protocol. Here, no downstream head is trained: we use the model's masked language modeling head to score mutations by comparing the log-likelihood of the mutated amino acid to that of the wild-type. We use the ProteinGym (Notin et al., 2023) DMS substitution benchmark, which compares predicted scores to experimental fitness scores for more than 2 million mutations over 217 assays. We used public figures for the ESM2, ESM-C, and ESM3 families of models.

**Analysis**  As shown in Figure 5, the structure-aligned models consistently outperform their unaligned counterparts across model families and sizes.

This consistent improvement in zero-shot fitness prediction is a strong indicator of an increase in biophysical understanding.

Protein language models rely on sequence co-evolution to predict fitness, essentially predicting fitness based on "what amino acid is common at this position?". In contrast, DMS assays measure experimental fitness, which is often dominated by protein stability. Our structure-aligned models overcome this limitation. By infusing inter-protein and intra-protein structural knowledge, they constrain the MLM head to favor mutations that are not just sequentially plausible but also structurally sound. A mutation that would destabilize the protein's fold is now correctly assigned a lower probability, leading to a stronger correlation with the experimental fitness data. Additional details about ProteinGym, including violin plots for all models, a pareto front analysis and head-to-head comparisons, are provided in Appendix D.3.

### 4.6  Pseudo-perplexity

To measure the impact of the structure alignment on the sequence-level knowledge of pLMs, we compute the pseudo-perplexity distributions of ESM2, AMPLIFY, and their structure-aligned variants as defined in Section 1.2.2 of Lin et al. (2022) using the validation set from Fournier et al. (2024). This set includes proteins with experimental evidence from reference proteomes derived from high-quality genomes across all three domains of life and is designed to reflect the natural protein distribution.

Figure 6 and Table 2 reveal that our structure alignment increases pseudo-perplexity on this validation set, which might appear as a trade-off induced, potentially by the distribution shift between UniProt and the PDB. However, the improved zero-shot DMS performance suggests a language modeling capability that is actually more aligned with the protein fitness landscape.

## 5 Discussion

### 5.1 Limitations

Our results support structure alignment as an inexpensive, useful post-training strategy, but they also clarify where the current formulation is limited. First, the alignment corpus is restricted to proteins with available structures. This is a practical constraint on the kinds of structural signals the model can receive during post-training, especially for proteins whose relevant biology is poorly captured by a single resolved structure, by static structural annotations, or by the current coverage of structure databases. More generally, our method uses a single structural view per protein during alignment, whereas disorder, conformational ensembles, environmental conditions, and post-translational modifications can make any single structure incomplete.

Second, our conclusions about broad functional transfer remain limited by the benchmarks themselves. Many functional-property tasks have small test sets, uncertain labels, or saturated baselines, making it difficult to distinguish a true absence of benefit from limited statistical resolution. We therefore interpret the functional results conservatively and use them primarily to delimit where the present form of structure alignment is most clearly useful.

### 5.2 Conclusion

In this work, we propose to enrich sequence-only pLMs with structural knowledge while preserving sequence-only inference. We incorporate structural insights from pre-trained pGNNs via a latent-level task that aligns residue representations across models, and we add a physical-level task that trains pLMs to predict structural tokens. We further introduce a residue loss selection module that emphasizes challenging yet reliable residue losses during post-training. Across ESM2 and AMPLIFY, this lightweight structure-alignment step improves the tasks most directly tied to structural signal, including contact prediction, secondary structure prediction, supervised stability prediction, and zero-shot DMS. At the same time, functional-property benchmarks often remain statistically inconclusive, and pseudo-perplexity increases on the validation set used to monitor sequence-level likelihood. Overall, these results support structure alignment as a practical way to inject structural knowledge into pLMs without making reliable structural input a requirement at inference, while also suggesting that structure and function should not be treated as interchangeable axes of representation quality.

**Availability Statement**   Model weights and both training and evaluation code (for the xTrimo and LLPS downstream tasks) will be made available upon publication. Evaluation for SaProt and ProteinGym relied on their publicly available repository.

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

# A  Related Work

## A.1  Structure Language Models

There are two main types of structure language models. The first requires explicit structural input, such as structure tokens (Su et al., 2024; Heinzinger et al., 2024; Li et al., 2024) or torsion angles (Frolova et al., 2024) or geometric graphs (Hartout et al., 2025). However, these models depend on potentially unreliable or inaccurate structural data (protein structures are generally modeled at cryogenic temperatures and fail to take into account the full conformational landscape), and protein structure databases like the PDB are much smaller than sequence-only databases. Additionally, many proteins lack a well-defined, rigid structure and contain disordered domains. All these approaches are not directly comparable to ours as their models require explicit structural inputs, whereas SaESM operates purely on sequences, a property we believe is important.

The second type only requires protein sequences as input and integrates structural insights during pre-training. For example, Zhang et al. (2024b) introduces a physical-level task for fold prediction, though it is somewhat coarse. Sun & Shen (2024) proposes several physical-level tasks, including secondary structure and distance map predictions, to incorporate structural knowledge into the pLM, while Ouyang-Zhang et al. (2024) focuses on structure token prediction. Peñaherrera & Koes (2024) uses a similar contrastive learning loss, but limits its focus to masked residues and does not utilize advanced pre-trained GNN models. Wang et al. (2025) and Su et al. (2025) primarily focus on latent embedding alignment and do not incorporate a physical-level task, which our ablation shows to be useful (Table 6). Furthermore, their contrastive learning is performed at the protein level, whereas our latent-level task operates at the residue level. We compare against S-PLM (Wang et al., 2025) as one of our baselines, but not against ProTek (Su et al., 2025) because of data leakage between their pretraining data and our downstream tasks. Note that the S-PLM post-training method adds approximately 100M parameters to ESM2, a more than 15% increase. You & Shen (2022) is limited to residue token prediction. Existing protein structure-alignment methods typically do not retain the original language-modeling head, which limits direct evaluation on zero-shot language-modeling tasks such as DMS. On the other hand, AlphaFold2 (Jumper et al., 2021) and ESMFold (Lin et al., 2023b) use sequence encoders, namely Evoformer and ESM2, followed by structure prediction modules. However, their focus is on structure prediction, and AlphaFold2 embeddings have been shown to be less effective than ESM2 embeddings for downstream tasks (Hu et al., 2022).

While recent studies have explored how to incorporate knowledge from pre-trained pLMs into pGNNs (Zheng & Li, 2024; Chen et al., 2023; Robinson et al., 2023), their focus is primarily on improving pGNNs rather than preserving and improving pLMs. Our work follows the complementary direction by introducing the latent-level task, thereby enriching pLMs with structural insights from pre-trained pGNNs. Finally, the idea of distilling structural information via some forms of contrastive learning between sequences is not new, as shown by (Bepler & Berger, 2021), which directly predicts contacts within a protein while simultaneously contrasting SCOP (Chandonia et al., 2017) information between protein pairs using a language modeling trunk.

## A.2  Data Selection

Data selection is a critical component in training protein models. AlphaFold2 (Jumper et al., 2021) filters proteins with a resolution higher than 9Å and excludes sequences where a single amino acid accounts for over 80% of the input sequence. Additionally, it samples protein chains based on length to rebalance distribution and cluster size to reduce redundancy, which risks deviating from the natural distribution shaped by evolutionary selection. ESM2 (Lin et al., 2023b) adopts comparable sampling strategies while AMPLIFY (Fournier et al., 2024) curates a validation set of proteins with experimental evidence at the protein or transcript level from reference proteomes derived from high-quality genomes across all three phylogenetic domains, aiming to better represent the natural protein distribution.

Data selection has also been extensively explored in natural language model pre-training, incorporating techniques such as filtering, heuristics, and domain-specific selection (Albalak et al., 2024). Our *residue loss selection* module is inspired by prior work (Lin et al., 2024), which uses excess loss to identify useful tokens in language pre-training. However, our approach differs significantly by operating at a finer granularity through

residue-level loss. Given the multi-loss structure of our framework, where each residue incurs three types of losses, we focus on those with the highest excess loss in each category. Crucially, our work is rooted in protein research rather than natural language, reflecting the unique challenges and requirements of protein modeling.

## B  Pseudo-perplexity Details

Table 2: Mean pseudo-perplexity on the validation set from Fournier et al. (2024). Paired t-tests compare each base model against its corresponding structure-aligned variant.

| Model | PPL | t-statistic | p-value |
|---|---|---|---|
| ESM2 (650M) | 5.89 | -8.65 | $5.50 \times 10^{-18}$ |
| SaESM2 (650M) | 6.38 | | |
| AMPLIFY (350M) | 4.58 | -9.30 | $1.55 \times 10^{-20}$ |
| SaAMPLIFY (350M) | 5.10 | | |

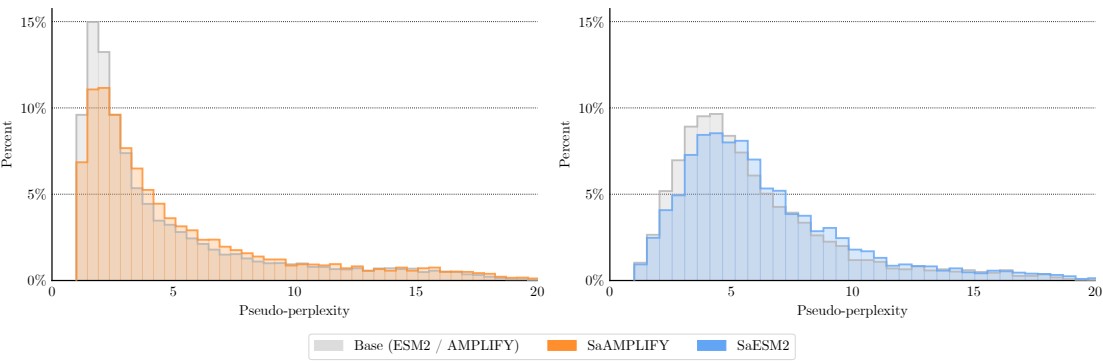

Figure 6: Pseudo-perplexity distributions on the validation set introduced by Fournier et al. (2024). Structure alignment increases pseudo-perplexity on this validation set.

## C  Residue Embedding Visualization

To qualitatively assess the effectiveness of our structure alignment technique, we visualize the residue embeddings from the final layer of ESM2 and AMPLIFY before and after alignment. Specifically, we analyze $1,000$ proteins from the Secondary Structure task, where each residue is color-coded based on its annotation to one of three secondary structure labels. We use UMAP (McInnes et al., 2018) to project high-dimensional data into a two-dimensional space with 50 nearest neighbors.

As shown in Figure 7, applying structure alignment improves the discrimination between secondary structures. In particular, the aligned embeddings (SaESM2 and SaAMPLIFY) exhibit clearer separation compared to their unaligned counterparts. Additionally, Figure 8 shows that amino acids sharing similar physical properties are located closer in the embedding space for aligned models compared to the unaligned baseline. Quantitative separability metrics are reported in Table 3.

Table 3: Quantitative evaluation of embedding separability. We report the **Silhouette Score** (measure of cluster cohesion/separation between -1 and 1, higher is better) and **k-NN Classification Accuracy** ($k = 20$) for residue type, grouped residue properties as in Figure 8, and Secondary Structure (Q3) labels. **Bold** values indicate the best performance within each model family.

| Model | Silhouette Score | | | k-NN Accuracy ($k = 20$) | | |
|---|---|---|---|---|---|---|
| | Amino Acid | Grouped AA | Sec. Str. (Q3) | Amino Acid | Grouped AA | Sec. Str. (Q3) |
| ESM2 | 0.023 | 0.005 | -0.001 | **0.981** | **0.983** | 0.772 |
| SaESM2 | **0.030** | **0.010** | **0.012** | 0.964 | 0.967 | **0.861** |
| AMPLIFY | 0.053 | **0.029** | -0.007 | 0.929 | 0.946 | 0.644 |
| SaAMPLIFY | **0.058** | 0.027 | **0.009** | **0.956** | **0.964** | **0.798** |

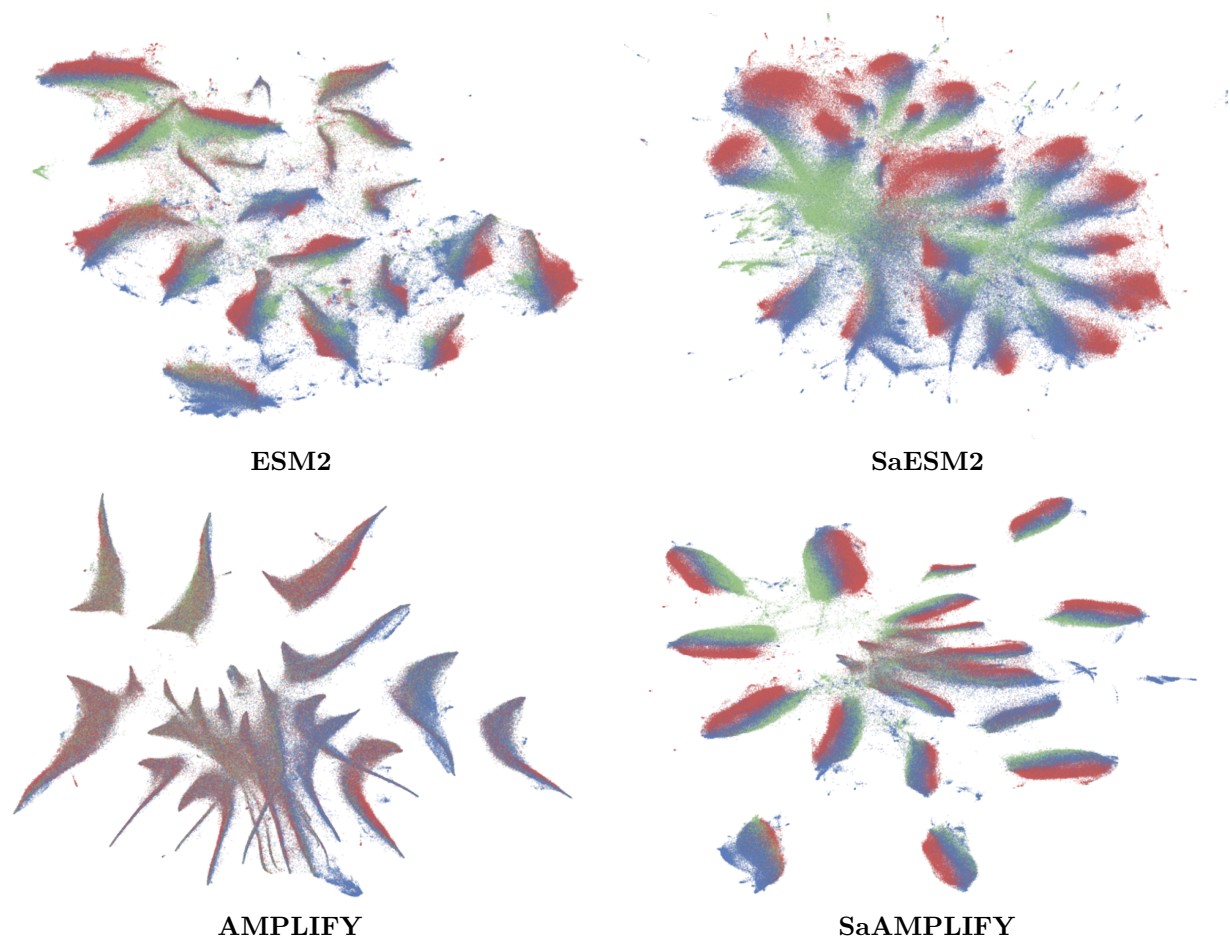

**ESM2**        **SaESM2**

**AMPLIFY**        **SaAMPLIFY**

Figure 7: Residue embeddings colored by secondary structure type across four models: ESM2, SaESM2, AMPLIFY, and SaAMPLIFY. Aligned models show cleaner separation between the three secondary-structure classes.

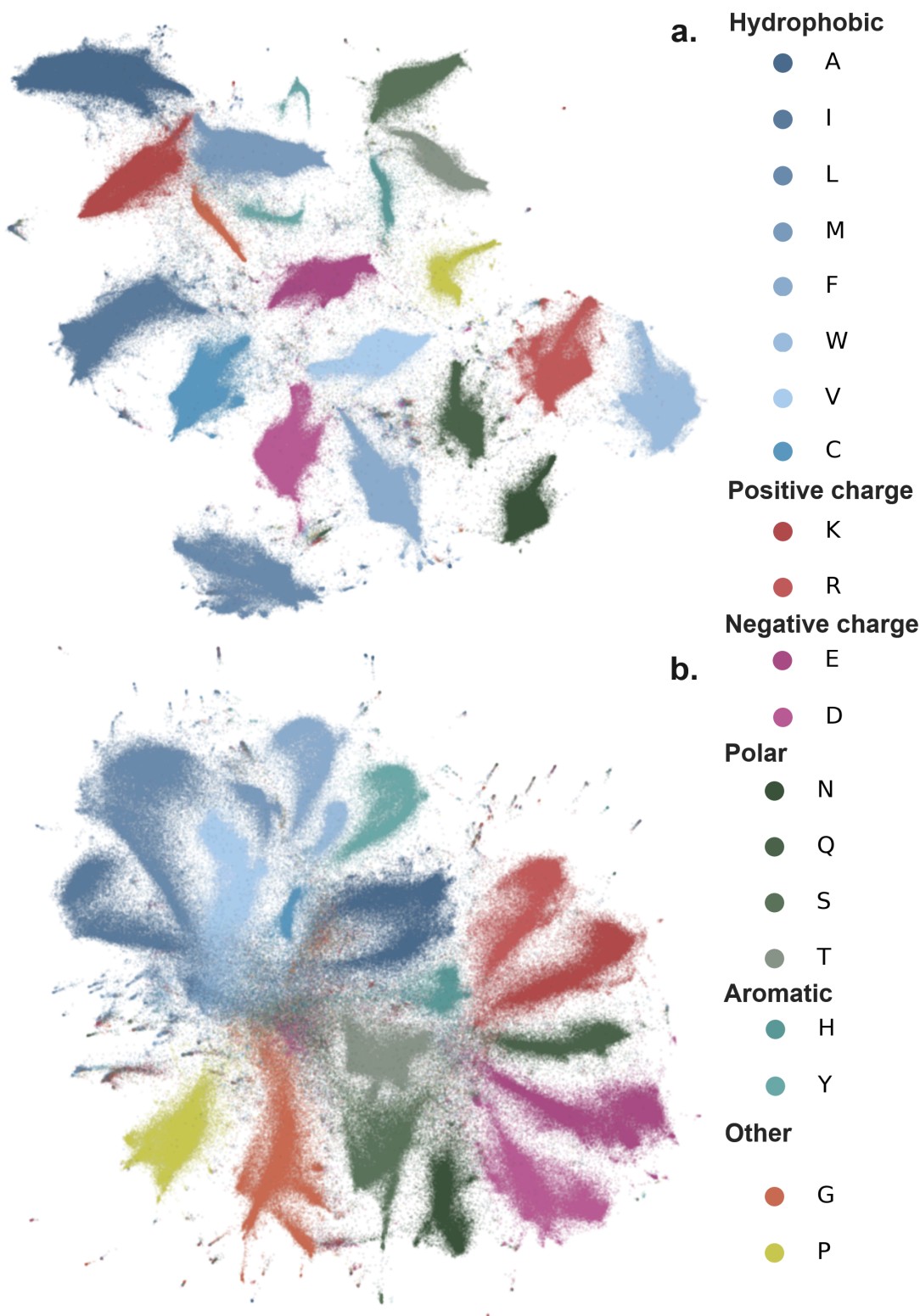

Figure 8: Residue embeddings colored by amino acid type for ESM2 (**a.**) and SaESM2 (**b.**). Amino acids with similar physical properties are colored with a gradient of the same color. Embeddings for SaESM2 clearly reveal a more physically coherent latent space.

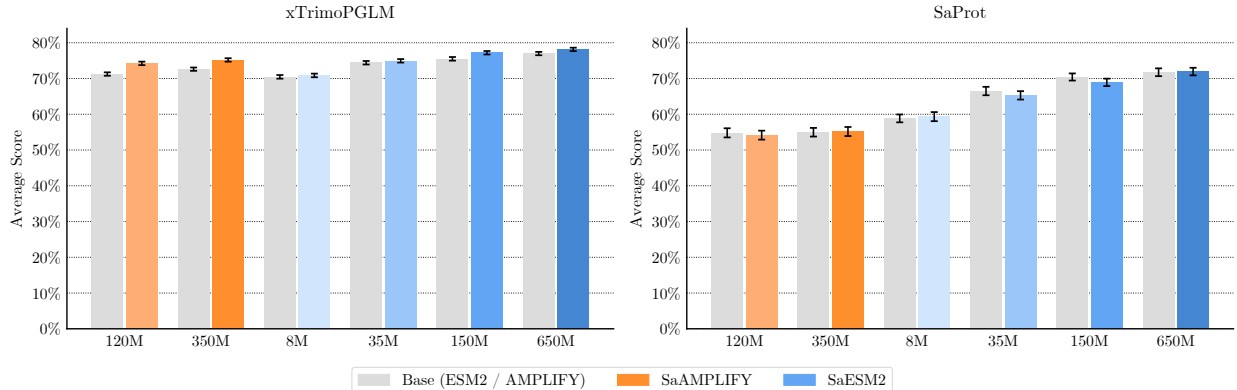

Figure 9: Average model performance compared with model size for different families of models over xTrimoPGLM (left) and SaProt (right).

## D Additional Results and Scaling Analysis

In this section, we provide additional results and visualizations for downstream property prediction benchmarks. These results separate the two regimes summarized in the main text: xTrimoPGLM contains several structure-proximal tasks where structure alignment improves performance, whereas the broader SaProt functional-property tasks mostly show overlapping confidence intervals and limited evidence for an additional benefit from alignment.

### D.1 Full Results on Downstream Property Prediction: xTrimoPGLM

In this section, we report the full results across model sizes for all evaluations considered from xTrimoPGLM. In Figure 9 (left), structure-aligned models outperform their baseline on average, with the largest gaps observed starting in the 100M parameter range. This average is driven mainly by the structure-proximal tasks described below rather than by uniform gains across every downstream endpoint. In Figure 10, we plot all per-task results.

Considering structure-based tasks that are not contact map (already discussed in Figure 3), we find consistent improvements in secondary structure prediction across all model sizes. For fold prediction, all SaESM2 models up to 150M parameters have significantly higher accuracy. For the 650M model, confidence intervals overlap while ESM2-S, for which the full backbone was trained in remote homology detection, remains best. SaAMPLIFY models outperform their baseline at all evaluated sizes.

For supervised mutation effect prediction, we report results across all model sizes for stability prediction, where SaESM2 achieves higher accuracy at both 150M and 650M.

Finally, for peptide-HLA MHC affinity, TCR-pMHC affinity, and enzyme catalytic efficiency, we do not observe a statistically reliable effect from structure alignment.

### D.2 Full Results on Downstream Property Prediction: SaProt

In this section, we report the full results across model sizes for all evaluations considered from SaProt. As shown in Figure 9 (right), these tasks mainly show scaling with the underlying sequence model size rather than a consistent additional gain from structure alignment.

The full results are displayed in Figure 12, with the largest model scales reported in Table 4. For most downstream evaluations, confidence intervals bootstrapped from the test set overlap substantially across models. This is the detailed counterpart to the main-text observation that broad functional benchmarks are heterogeneous: some endpoints improve with stronger base sequence models, but the structure-alignment variants evaluated here do not separate clearly from their baselines. Possible explanations include saturation

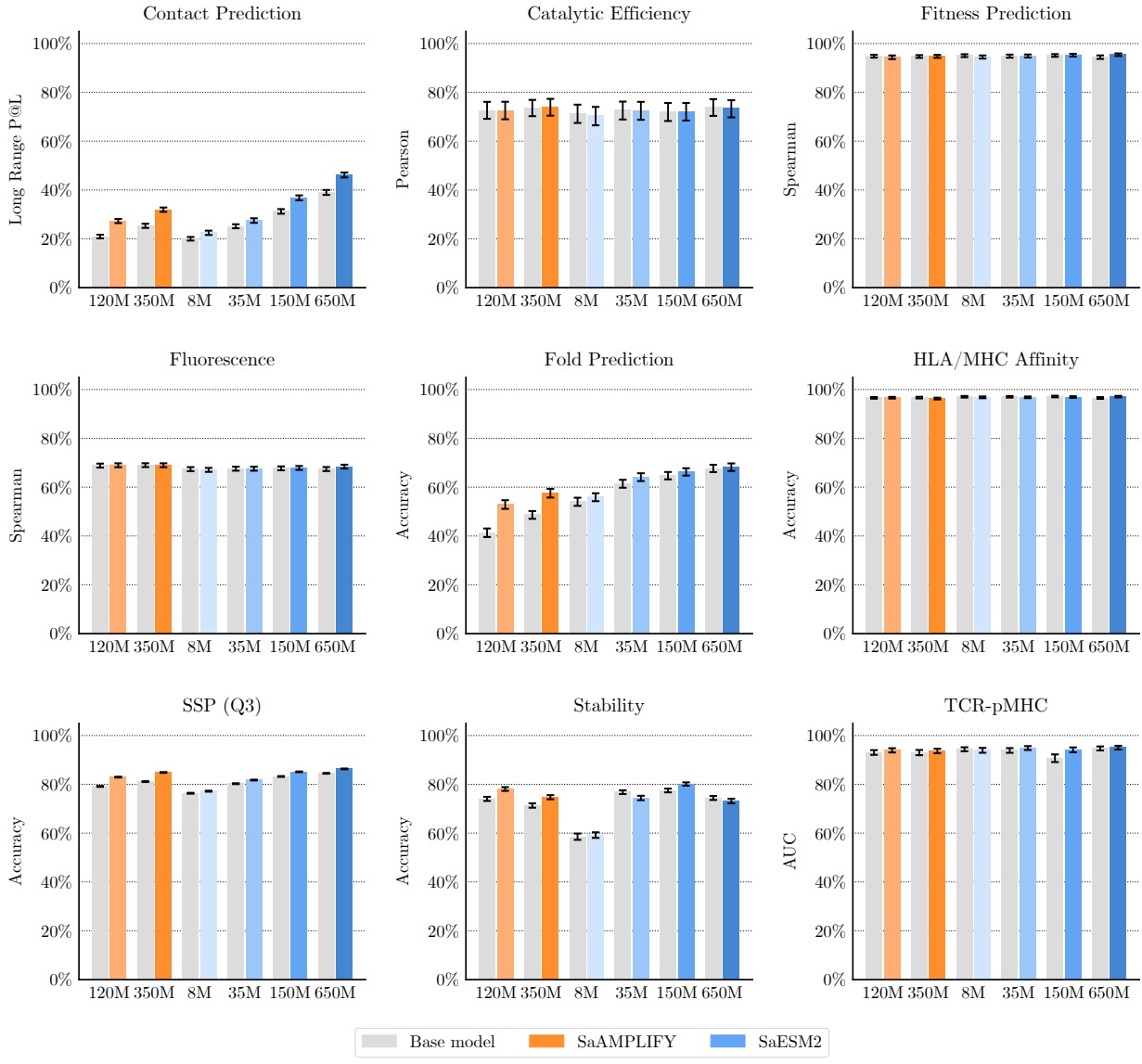

Figure 10: Model performance compared with model size for different families of models over every xTri-moPGLM task.

for these particular supervised splits, noisy training labels, or test sets that are too small relative to task noise to resolve modest representation differences.

### D.3 Additional Results on ProteinGym

In Figure 15, we present full violin plots for all models involved in our comparison, including public reference models such as larger ESM2 variants. In Figure 14, we focus on the scaling and Pareto-front comparison among AMPLIFY/SaAMPLIFY, ESM2/SaESM2 up to 650M parameters, and public ESM-C and ESM3 reference points. Finally, Figure 16 compares the performance of our models on every assay for the original model and the structure-aligned one. For smaller-sized models, our method appears to reduce the number of assays in which our fitness predictions are anti-correlated with the ground truth. For every model family and size, the improvements seem to be due in equal part to: (i) a slight improvement on average on most of the assays, (ii) some assays where Spearman correlations are substantially improved. A closer inspection of these

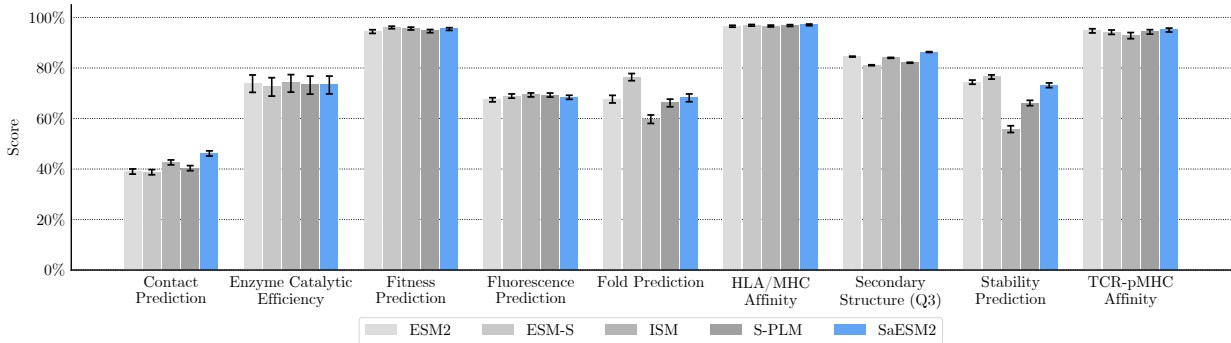

Figure 11: Per-task performance on the xTrimoPGLM benchmark for 650M-scale models, with 95% bootstrap confidence intervals.

Table 4: Results on functional property prediction. Values are Metric [95% Confidence Interval]. Within each model family (ESM2-based and AMPLIFY-based), the **best-performing** model is bolded. Models in *italics* have a mean score that falls within the 95% CI of the best model in their family. SaESM2 falls within the confidence intervals for all tasks where it's not best, except for GO Cellular Component and Human PPI. Confidence intervals for models marked with † and tasks marked with ‡ are statistical upper bounds.

| Model | EC
Fmax (↑) | GO (BP)
Fmax (↑) | GO (CC)
Fmax (↑) | GO (MF)
Fmax (↑) | Thermostability
Sp. (↑) |
|---|---|---|---|---|---|
| ESM2 | 0.855 [0.841, 0.869] | *0.477* [0.467, 0.487] | *0.484* [0.474, 0.493] | *0.672* [0.661, 0.681] | **0.712** [0.573, 0.765] |
| ESM2-S† | 0.861 [0.844, 0.878] | *0.479* [0.462, 0.496] | 0.458 [0.441, 0.475] | **0.673** [0.657, 0.689] | *0.683* [0.658, 0.708] |
| ISM† | *0.872* [0.856, 0.888] | *0.471* [0.454, 0.488] | **0.497** [0.480, 0.513] | *0.666* [0.650, 0.682] | *0.695* [0.671, 0.719] |
| S-PLM | **0.878** [0.866, 0.892] | **0.480** [0.472, 0.491] | 0.445 [0.435, 0.455] | *0.671* [0.660, 0.682] | *0.704* [0.590, 0.766] |
| SaESM2 (ours) | *0.868* [0.855, 0.882] | *0.479* [0.470, 0.489] | 0.462 [0.452, 0.473] | *0.663* [0.653, 0.674] | *0.693* [0.570, 0.756] |
| AMPLIFY | **0.501** [0.480, 0.525] | **0.271** [0.263, 0.279] | 0.322 [0.311, 0.332] | *0.378* [0.366, 0.393] | **0.614** [0.430, 0.640] |
| SaAMPLIFY (ours) | *0.486* [0.464, 0.508] | 0.257 [0.250, 0.266] | **0.348** [0.342, 0.358] | **0.389** [0.376, 0.401] | *0.596* [0.420, 0.641] |

| Model | DeepLoc (Subcell.)‡
Acc (↑) | DeepLoc (Binary)‡
Acc (↑) | HumanPPI‡
Acc (↑) | Metal Bind‡
Acc (↑) |
|---|---|---|---|---|
| ESM2 | *0.839* [0.825, 0.852] | *0.931* [0.919, 0.942] | 0.783 [0.722, 0.836] | 0.705 [0.671, 0.740] |
| ESM2-S† | 0.828 [0.814, 0.842] | **0.934** [0.923, 0.945] | *0.826* [0.771, 0.875] | 0.711 [0.677, 0.745] |
| ISM† | 0.826 [0.812, 0.840] | *0.923* [0.910, 0.935] | *0.815* [0.759, 0.866] | 0.699 [0.665, 0.734] |
| S-PLM | **0.847** [0.834, 0.860] | *0.930* [0.917, 0.942] | **0.853** [0.799, 0.902] | 0.696 [0.661, 0.731] |
| SaESM2 (ours) | *0.840* [0.826, 0.853] | *0.933* [0.921, 0.944] | 0.777 [0.716, 0.831] | **0.759** [0.726, 0.790] |
| AMPLIFY | **0.689** [0.672, 0.706] | 0.861 [0.845, 0.877] | *0.690* [0.623, 0.756] | **0.621** [0.584, 0.657] |
| SaAMPLIFY (ours) | *0.674* [0.656, 0.691] | **0.881** [0.865, 0.895] | **0.734** [0.669, 0.796] | *0.601* [0.564, 0.637] |

assays over the three standard ProteinGym assay metadata reveals no correlation between the assay type and the improvement from our method.

In the scaling view, the structure-aligned models move their corresponding base families upward at essentially fixed model size. This is the comparison used for the Pareto-front statement: SaESM2 and SaAMPLIFY should be interpreted relative to their unaligned counterparts and to the ESM-C/ESM3 external reference curve, not as a claim about every public model shown in the violin plot.

### D.4 Stability of our Post-Training Method: Analysis of the Downstream Performance over 3 Seeds

To validate the stability of our structure-alignment procedure, we performed post-training using 3 different random seeds and measured the standard deviation of the final test metric. In Table 5, we compare this post-training variance against the inherent uncertainty of the benchmarks themselves, represented by the 95%

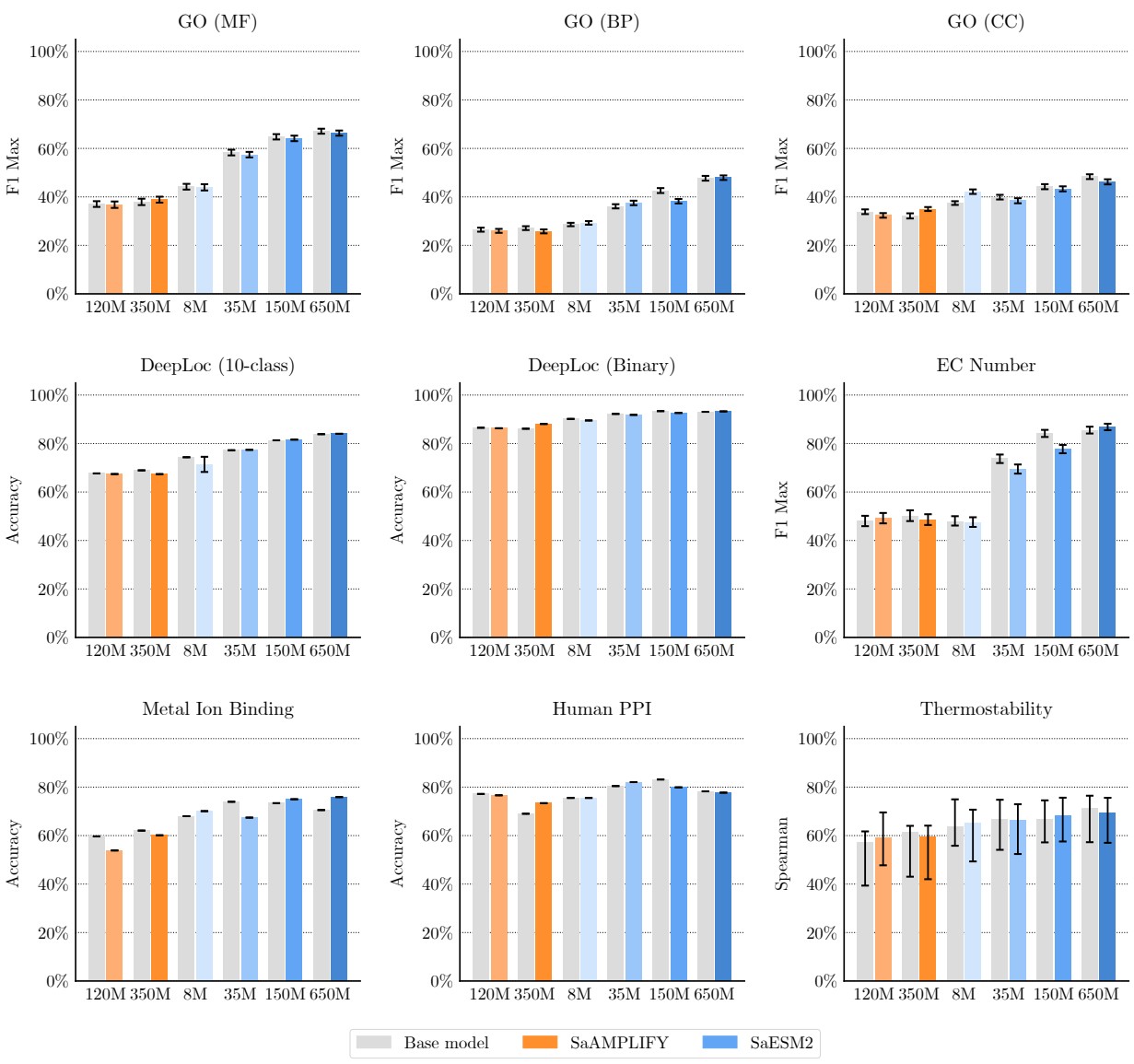

Figure 12: Model performance compared with model size for different families of models over every SaProt task. Confidence intervals for models marked with † and tasks marked with ‡ are statistical upper bounds on confidence intervals. For other tasks and models, confidence intervals are computed via bootstrapping on the test set.

confidence interval (CI) size derived from bootstrapping the test set. For 8 out of the 9 tasks, the standard deviation from our post-training seeds is smaller than the test set CI, often by an order of magnitude (e.g., 0.0027 vs. 0.0108 for Fitness Prediction). This result is crucial, as it indicates that our method is robust and that the observed variance in benchmark scores is driven by the test set's composition rather than the stochasticity of our alignment process. We note that Stability Prediction is an exception, showing higher seed variance than test set uncertainty. After further investigation, this sensitivity is mainly due to the fine-tuning pipeline, as we found that our Stability Prediction results are particularly sensitive to hyperparameter choices.

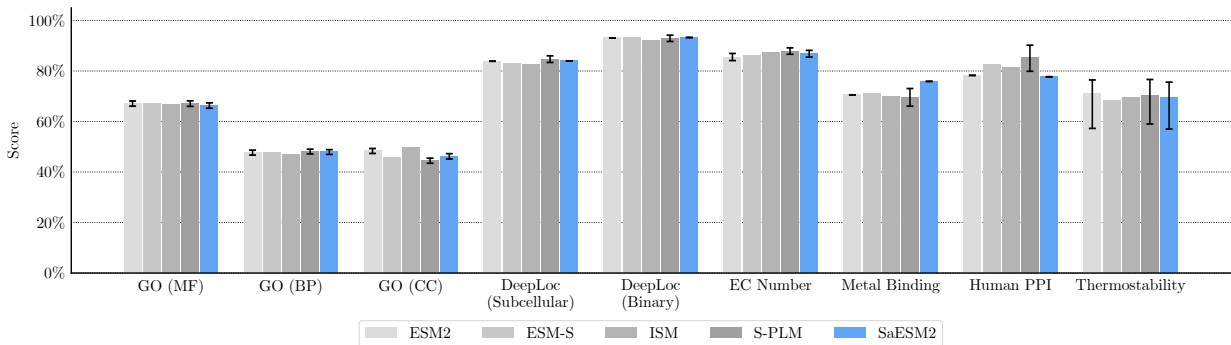

Figure 13: Per-task performance on the SaProt benchmark for 650M-scale models, with 95% bootstrap confidence intervals.

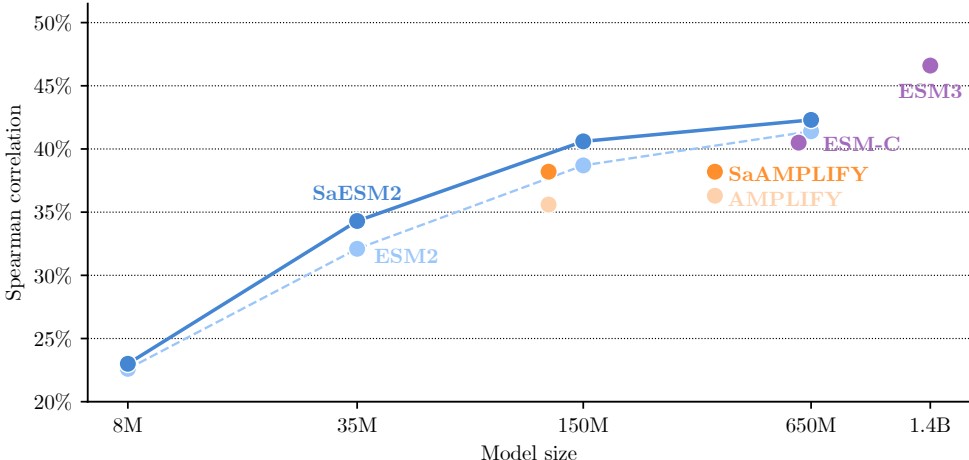

Figure 14: Average Spearman correlation on the 217-assay ProteinGym DMS substitution benchmark across model sizes and families. The scaling comparison includes AMPLIFY/SaAMPLIFY, ESM2/SaESM2 up to 650M parameters, and public ESM-C and ESM3 values as external reference points. Structure-aligned models (SaESM2, SaAMPLIFY) consistently outperform their unaligned counterparts across the evaluated sizes.

### D.5 Ablation Studies

We conduct extensive ablation studies on three tasks covering structure (Contact), mutation effect (Fluorescence), and property (Metal Bind) to evaluate the contribution of each design component.

**Dual-Task Framework** Our default setup employs a weighted combination of three losses: masked language modeling, latent-level, and physical-level, with weights (1, 0.5, 0.5), respectively. To assess the impact of each component, we experiment with the following configurations:

- *w/o latent*: Remove the latent-level loss, using weights (1, 0, 0.5).

- *w/o physical*: Remove the physical-level loss, using weights (1, 0.5, 0).

- *w/o dual*: Exclude both auxiliary losses, i.e., MLM fine-tuning on PDB, using weights (1, 0, 0).

As shown in Table 6, removing either structural objective reduces performance across the three ablation tasks, confirming that the latent-level and physical-level losses are complementary. The *w/o dual* setting, which corresponds to MLM-only post-training on PDB sequences, also performs worse than the full model and even below the original ESM2 baseline on contact prediction. This is consistent with the distribution shift

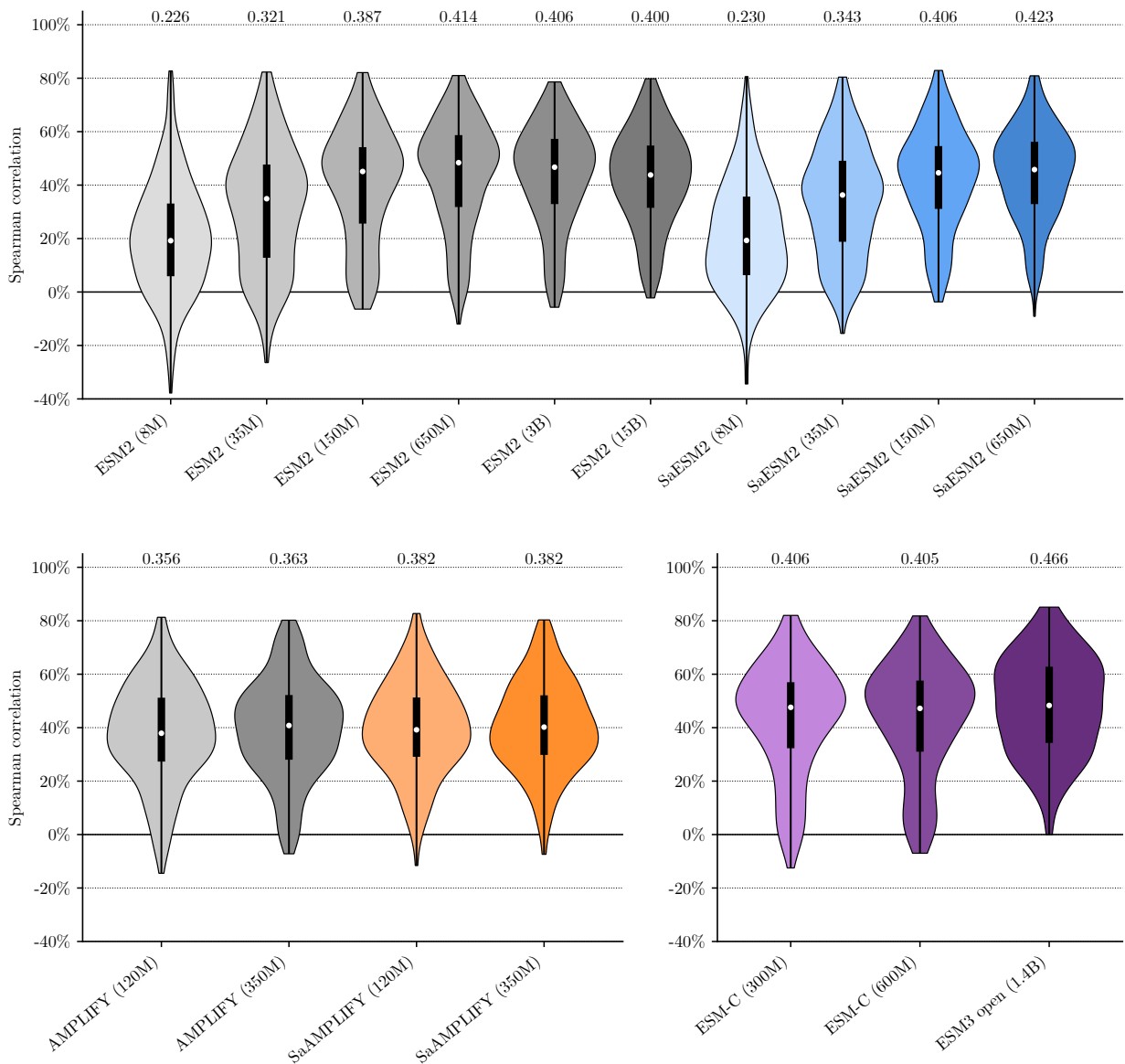

Figure 15: Violin plots of the distribution of assay Spearman correlations for all models evaluated on ProteinGym. The solid black line at 0 represents the expected correlation of a random model.

introduced by PDB-only post-training and supports the role of the structural objectives in the alignment step. In these point estimates, the *w/o latent* setting performs worse than *w/o physical*, suggesting that the latent-level task contributes more to the considered downstream tasks than the physical-level task.

**Residue Loss Selection**   We compare our *residue-level selection* module with two alternative strategies that do not rely on reference models, instead selecting residues based solely on their individual loss values:

- *loss-large*: Select residues with high losses, assuming they offer greater learning potential.

- *loss-small*: Select residues with low losses, assuming they are cleaner and more accurate.

For comparison, we also include a *full* strategy that uses all residue losses without any selection.

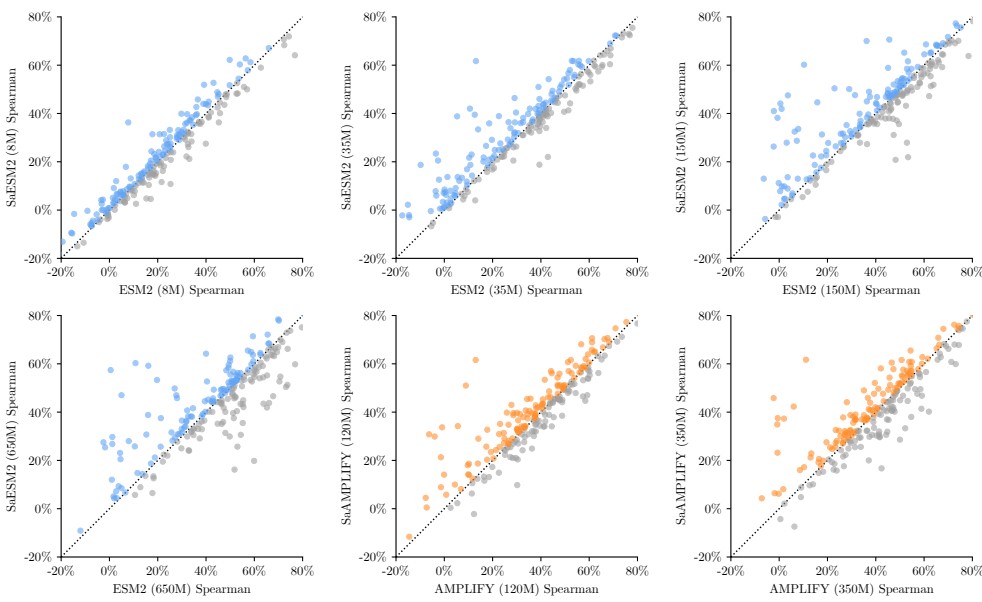

Figure 16: Head-to-head comparison of model performance on ProteinGym. Each point represents a single deep mutational scanning (DMS) assay. **(Top)** SaESM2 (y-axis) vs. ESM2 (x-axis). **(Bottom)** SaAMPLIFY (y-axis) vs. AMPLIFY (x-axis). Points above the $y = x$ diagonal (dashed line) indicate an improvement over the assay from our structure alignment.

Table 5: Robustness of the post-training procedure. We compare the standard deviation of the final test metric across 3 independent post-training seeds (**Std. Dev. (3 Seeds)**) against the 95% confidence interval (CI) size computed via bootstrapping on the test set (**Test Set CI Size**). The low standard deviation across seeds demonstrates the high stability of our alignment method.

| Task | Std. Dev. (3 Seeds) | Test Set CI Size |
|---|---|---|
| Contact Prediction CASP16 (Long Range P@L) | 0.0031 | 0.0204 |
| Enzyme Catalytic Efficiency (Pearson Correlation) | 0.0022 | 0.0705 |
| Fitness Prediction (Spearman Correlation) | 0.0027 | 0.0108 |
| Fluorescence Prediction (Spearman Correlation) | 0.0084 | 0.0160 |
| Fold Prediction (Accuracy) | 0.0026 | 0.0309 |
| Peptide-HLA MHC Affinity (AUC) | 0.0027 | 0.0066 |
| Secondary Structure Prediction (Accuracy) | 0.0013 | 0.0032 |
| Stability Prediction (Accuracy) | 0.0766 | 0.0185 |
| TCR-pMHC Affinity (AUC) | 0.0022 | 0.0154 |

As shown in Table 7, alternative selection strategies have lower point estimates across all tasks, indicating that the *residue loss selection* module provides a modest but consistent gain. Its impact is smaller than that of the *dual-task framework*, likely due to the already high quality of the protein structures used and the extensive pre-training of base pLMs. We further visualize the validation loss curves for different loss selection strategies in §E, which supports the same trend.

**Structure Embedding**  We further ablate the structure embeddings used in the latent-level task. In addition to our default GearNet embeddings (Zhang et al., 2023), we explore embeddings from the AlphaFold2 Evoformer model (Jumper et al., 2021), denoted as *AF2*. Specifically, we provide the protein structure as a template and perform only one Evoformer cycle to extract the embeddings, thereby reducing computational cost.

Table 6: Ablations on dual-task framework.

| | Contact on the trRosetta split | Fluorescence | Metal Bind |
|---|---|---|---|
| | P@L/5 (↑) | Spearman (↑) | Acc% (↑) |
| **SaESM2** (*all tasks and losses*) | **61.0** | **0.695** | **72.3** |
| *w/o latent* | 53.7 (−12.0%) | 0.689 (−0.9%) | 69.5 (−3.8%) |
| *w/o physical* | 59.1 (−3.1%) | 0.691 (−0.6%) | 71.0 (−1.8%) |
| *w/o dual* | 51.4 (−15.7%) | 0.686 (−1.3%) | 67.1 (−7.2%) |
| ESM2 (*baseline*) | 54.1 (−11.3%) | 0.687 (−1.2%) | 70.8 (−2.1%) |

Table 7: Ablations on residue loss selection.

| | Contact on the trRosetta split | Fluorescence | Metal Bind |
|---|---|---|---|
| | P@L/5 (↑) | Spearman (↑) | Acc% (↑) |
| **SaESM2** (*residue-loss selection*) | **61.0** | **0.695** | **72.3** |
| *loss-large* | 60.6 (−0.7%) | 0.693 (−0.3%) | 71.3 (−1.4%) |
| *loss-small* | 59.4 (−2.6%) | 0.691 (−0.6%) | 71.0 (−1.8%) |
| *full* | 60.3 (−1.1%) | 0.690 (−0.7%) | 71.1 (−1.7%) |
| ESM2 (*baseline*) | 54.1 (−11.3%) | 0.687 (−1.2%) | 70.8 (−2.1%) |

Table 8: Ablations on structure embedding.

| | Contact on the trRosetta split | Fluorescence | Metal Bind |
|---|---|---|---|
| | P@L/5 (↑) | Spearman (↑) | Acc% (↑) |
| **SaESM2** (*GearNet embeddings*) | **61.0** | **0.695** | **72.3** |
| *AF2* | 48.4 (−20.7%) | **0.695** (−0.0%) | 69.0 (−4.6%) |
| ESM2 (*baseline*) | 54.1 (−11.3%) | 0.687 (−1.2%) | 70.8 (−2.1%) |

Table 9: Ablations on structure token.

| | Contact on the trRosetta split | Fluorescence | Metal Bind |
|---|---|---|---|
| | P@L/5 (↑) | Spearman (↑) | Acc% (↑) |
| **SaESM2** (*foldseek struct. tokens*) | 61.0 | **0.695** | **72.3** |
| *protoken* | 60.8 (−0.3%) | **0.695** (+0.0%) | 71.9 (−0.6%) |
| *aido* | **61.9** (+1.5%) | **0.695** (+0.0%) | 70.5 (−2.5%) |
| ESM2 | 54.1 (−11.3%) | 0.687 (−1.2%) | 70.8 (−2.1%) |

As shown in Table 8, aligning with GearNet embeddings outperforms aligning with AlphaFold2 embeddings on both Contact Prediction (trRosetta split) and Metal Bind tasks. We also observed a degradation of our method when aligning to AF2 embeddings compared to the baseline ESM2 model without structural alignment. This observation is consistent with the findings of Hu et al. (2022), which suggest that embeddings from the AF2 may not be well-suited for some downstream tasks.

**Structure Token**  We further ablate the structure token used in the physical-level task. Our approach is based on *foldseek* structure tokens (van Kempen et al., 2022) and we explore *protoken* (Lin et al., 2023a) and *aido* (Zhang et al., 2024a), both of which employ a larger codebook size (512 compared to 20 for *foldseek*). We do not compare against the *ESM3* structure token (Hayes et al., 2024) due to its strict commercial license.

As shown in Table 9, *aido* has the highest point estimate on the contact prediction task, likely due to its finer-grained structural representation that injects richer structural insights into the pLM. *Protoken* performs

slightly worse despite its larger codebook, likely due to *protoken* encoding global dependencies instead of emphasizing local neighborhoods like *foldseek*, which aligns more closely with our structure alignment approach. This observation is consistent with that of Zhang et al. (2024a). For the property prediction task Metal Bind, *foldseek* performs best, supporting the importance of local structure. All three tokens perform similarly on the fluorescence prediction task.

# E    Loss Curve Analysis of Residue Loss Selection

To assess the effectiveness of our proposed *residue loss selection* module, we analyze validation loss curves across four strategies: ours, loss large, loss small, and full. These are shown in Figure 17 (overall loss), Figure 18 (MLM loss), Figure 19 (latent-level loss), and Figure 20 (physical-level loss). Recall that the overall loss is defined as:

$$\mathcal{L}_{\text{overall}} = \mathcal{L}_{\text{mlm}} + 0.5\mathcal{L}_{\text{latent}} + 0.5\mathcal{L}_{\text{physical}}. \tag{11}$$

As seen in Figure 17, our strategy consistently achieves the lowest overall loss, demonstrating superior training effectiveness and efficiency. Figure 19 shows that the primary reduction comes from the latent-level loss, indicating that our method successfully identifies informative and challenging latent-level residue losses to enhance learning. In contrast, Figure 20 shows negligible differences in physical-level loss across most strategies, except for loss small. We attribute this to the limited Foldseek codebook size (20), which provides only coarse structural information, thereby reducing the potential benefit of residue loss selection at this level. Notably, the loss small strategy results in high physical-level loss, likely due to its focus on easy-to-learn residues, which fail to contribute meaningful structural insights to the pLMs. We further experiment with joint training on both the training and validation sets. However, this led to degraded downstream performance, likely due to overfitting on the validation set.

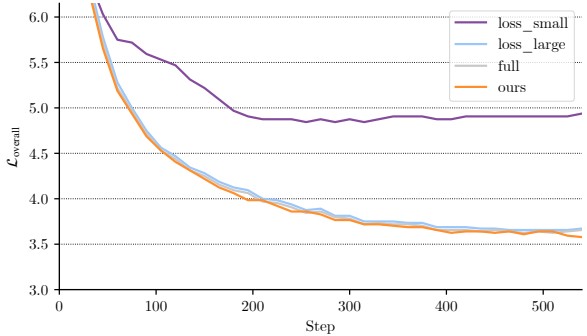

Figure 17: Overall loss.

Figure 18: MLM loss.

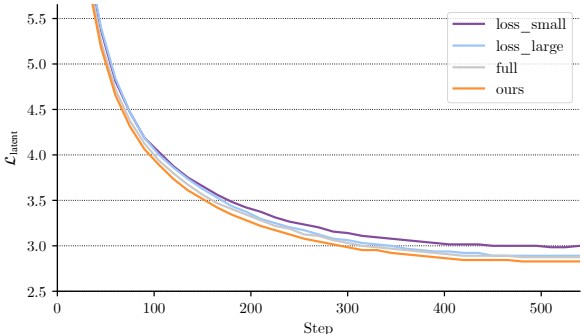

Figure 19: Latent-level loss.

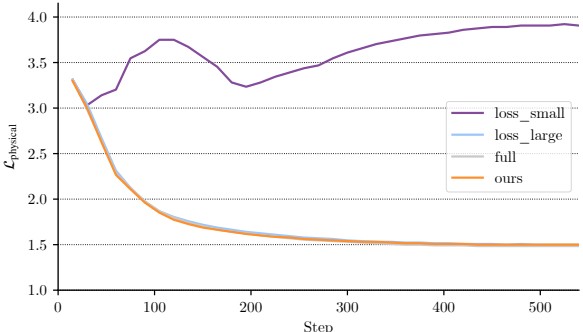

Figure 20: Physical-level loss.

