# OpenReview forum: "Structure-Aligned Protein Language Model"
_TMLR — Under review for TMLR_

### Review · Reviewer_EEo4 · 2026-07-04

**Summary Of Contributions:**

This paper introduces "structure alignment", a lightweight post-training procedure that injects structural knowledge into sequence-only protein language models (pLMs) such as ESM2 and AMPLIFY, while keeping inference purely sequence-based (no structure input required at test time). The method has three parts: (i) a residue-level contrastive "latent" task that aligns pLM residue embeddings with those of a frozen pre-trained protein GNN (GearNet), injecting inter-protein structural signal; (ii) a "physical" task that trains the pLM to predict discrete structure tokens (Foldseek), injecting intra-protein signal; and (iii) a residue loss selection module that uses the excess loss relative to a smaller reference model trained on high-quality PDB structures (resolution < 2.0 Angstrom, R-free < 0.20) to prioritize challenging yet reliable residues. The original masked language modeling head is retained, which uniquely enables direct zero-shot evaluation on deep mutational scanning (DMS).

The empirical study is broad and, importantly, honest about scope. Gains are concentrated on tasks tightly coupled to structure and biophysics (contact prediction on CASP16 and trRosetta, secondary structure, supervised stability, and zero-shot DMS over 217 ProteinGym assays), while the authors explicitly report that broader functional benchmarks (9 xTrimoPGLM and 9 SaProt tasks) are largely statistically inconclusive or saturated. A distinct and scientifically valuable framing contribution is the disorder-stratified LLPS condensate analysis, which shows that structure-conditioned models (SaProt, ESM-GearNet on predicted structures) degrade as intrinsic disorder increases, whereas sequence-only and structure-aligned models remain robust.

Key strengths:
- Strong, current, and fairly handled baselines (ESM2-S, ISM, S-PLM), with explicit accounting of confounds (S-PLM adds ~100M parameters) and data leakage (ProTek excluded; training set deduplicated against CASP16).
- Rigorous statistical reporting: 95% bootstrap confidence intervals, full assay-level ProteinGym distributions, and a dedicated 3-seed post-training stability analysis.
- Genuine, biologically grounded insight: structure and function appear to be partly separable axes of representation quality, and structure conditioning is a liability for disordered proteins.
- Ten ablations isolating each design component, plus structure-embedding and structure-token substitutions.
- Practical value: a compute-light recipe (under 6 hours on 8xH100 for the 650M model) that preserves sequence-only inference.

Key weaknesses:
- The payoff is heterogeneous; across 27 functional comparisons the aligned models beat baselines only 14 times, so clear wins are confined to structure-proximal and fitness tasks.
- The flagship result (a 59% relative increase in contact P@L/5 on CASP16) rests on a small, partly redundant split (51 chains from 13 complexes) with wide confidence intervals.
- The core mechanism is a well-executed recombination of existing ideas (CLIP-style contrastive alignment, structure-token prediction, excess-loss data selection); the residue loss selection module contributes only marginally.
- Some interpretive claims (e.g., that DMS gains indicate increased "biophysical understanding") are stated more strongly than the evidence supports, and the increased pseudo-perplexity is reframed favorably rather than analyzed as a clear trade-off.

**Additional Comments:**

This is a rigorous and refreshingly honest paper. The willingness to report that broad functional benchmarks show no reliable improvement, and to frame this as evidence that structure and function are partly separable, is a strength rather than a weakness and should be preserved in revision rather than downplayed. The main opportunities for improvement are in calibrating the few overstated interpretive claims and in shoring up the small-split and reproducibility details; none of these undermine the core contribution, and addressing them would make the paper a solid and durable reference for structure-informed protein language modeling.

**Audience:**

Yes

**Audience Explanation:**

Protein language models are a major and active area of interest for the TMLR audience, spanning representation learning, contrastive methods, multimodal alignment, and applications in computational biology and drug discovery. This paper speaks to several of these communities at once.

The finding most likely to be of broad interest is that structure and function behave as partly separable axes of representation quality: reinforcing structural signal reliably helps geometry- and fitness-related tasks (contact, secondary structure, stability, zero-shot DMS) but does not uniformly transfer to broader functional annotation. This is a useful, somewhat cautionary result for anyone assuming that "more structure knowledge" monotonically improves pLMs. The complementary observation that structure-conditioned models degrade on intrinsically disordered proteins, while sequence-only and structure-aligned models remain robust, is practically important because disordered regions constitute a large fraction of the human proteome and are exactly where predicted structures are least reliable.

Beyond the biology, the methodological pattern (aligning a sequence model to a frozen pre-trained encoder of another modality at the residue/token level while retaining the original language-modeling head) is transferable and of general interest to researchers working on knowledge distillation and cross-modal alignment. The unusually careful evaluation, including the honest reporting of negative and inconclusive results, also provides a good template for benchmarking in this space.

**Broader Impact Concerns:**

I have no significant ethical concerns that would block acceptance. The work is methodological, uses public protein data (PDB, OpenFold, DrLLPS, MobiDB, ProteinGym, xTrimoPGLM, SaProt benchmarks), and does not involve human subjects or personal data.

The paper does not include a Broader Impact Statement. Given that improved protein fitness and stability prediction sits within drug-discovery and protein-engineering pipelines, it would be appropriate to add a brief statement acknowledging dual-use considerations (for example, that improved fitness/stability modeling could in principle be applied to engineering harmful proteins) alongside the clear positive applications, and confirming the licenses and terms of use for the datasets and pre-trained models employed (the paper already notes it avoided ESM3 and ProTek due to licensing and leakage, which is good practice). A short paragraph would be sufficient; I do not consider the absence of such a statement to be a critical flaw.

**Claims And Evidence:**

Yes

**Claims Explanation:**

Overall the central claims are well supported, and the paper is notable for calibrating its claims to its evidence rather than overselling.

The main claim, that structure alignment as a post-training step improves performance on structure-proximal and fitness tasks while preserving sequence-only inference, is convincingly demonstrated in Table 1 and Figures 3 and 5, with bootstrap confidence intervals and consistent trends across model sizes (150M to 650M) and two model families (ESM2, AMPLIFY). The zero-shot DMS improvements across 217 ProteinGym assays (Figure 5, Appendix D.3) are the strongest and most reproducible piece of evidence. The ablations (Appendix D.5, Tables 6 to 9) adequately isolate the contributions of the latent task, the physical task, and the loss selection module, and the 3-seed stability analysis (Table 5) shows that post-training variance is typically well below test-set uncertainty.

The claim that structure-conditioned models fail under disorder while sequence-only and structure-aligned models do not (Figure 2) is supported and represents the paper's most scientific finding. One caveat is that the structure-conditioned baselines rely on ESMFold-predicted structures for the disordered regions, so part of the observed degradation could stem from folding errors on IDRs rather than from structure conditioning per se; the authors would strengthen this claim by disentangling the two.

Two places warrant caution rather than disbelief. First, the headline "59% increase in P@L/5" is a relative gain on a small CASP16 split (51 chains from 13 complexes) with wide and overlapping confidence intervals, so the magnitude should be interpreted carefully even though the direction is consistent with the larger trRosetta split. Second, the interpretive statement in Section 4.5 that DMS gains are "a strong indicator of an increase in biophysical understanding", together with the favorable reframing of the pseudo-perplexity increase (Table 2) as better fitness-landscape alignment, goes beyond what the current experiments directly establish. These are matters of framing and additional controls, not of unsupported core results. The honest reporting of inconclusive functional-benchmark results (Section 4.4.2, Figure 4) actually increases confidence in the overall evidence.

**Requested Changes:**

Critical to securing my recommendation for acceptance:

1. Strengthen or appropriately qualify the flagship contact-prediction claim. The CASP16 split (51 chains from 13 complexes, with acknowledged chain-level redundancy) is small and yields wide confidence intervals. Please either report a larger, non-redundant structural evaluation, aggregate at the complex level, or explicitly foreground the small-N caveat wherever the "59%" figure appears (including the abstract). This is important because the number is currently a headline result.

2. Provide the deduplication and data-split details for all downstream benchmarks, not only LLPS. Leakage control is load-bearing for essentially every reported gain. The paper states sequences were deposited no later than December 2021 and that the training set is deduplicated against CASP16, but the exact thresholds and procedures for the xTrimoPGLM, SaProt, and ProteinGym evaluations are under-specified. A short table summarizing identity/coverage thresholds per benchmark would resolve this.

3. Temper or substantiate the interpretive claims around DMS and pseudo-perplexity. The statement that DMS gains indicate increased "biophysical understanding" (Section 4.5) and the framing of the pseudo-perplexity increase (Table 2, Section 4.6) as improved fitness-landscape alignment should either be softened to hypotheses or supported with a direct control (for example, a PDB-matched perplexity comparison to separate distribution shift from genuine likelihood realignment).

Would strengthen the work but are not strictly required:

4. Disentangle the disorder result (Figure 2) from ESMFold prediction error. Because the structure-conditioned baselines use predicted structures on IDRs, an experiment using experimental or AlphaFold2 structures where available (or reporting predicted-structure confidence such as pLDDT stratification) would clarify how much of the degradation is intrinsic to structure conditioning versus an artifact of folding errors.

5. Justify the residue loss selection module against its overhead. Its gains are small (roughly 0.3% to 2.6% in Table 7), yet it requires training separate reference models. Please report the compute cost of the reference-model pipeline and clarify when the module is worth using versus simply using all residue losses.

6. Clarify reproducibility. Code and weights are promised upon publication. Please confirm the release plan and, in the meantime, add missing details needed for reimplementation (reference-model training configuration, projection dimension D, temperature initialization, and the precise construction of the high-quality reference set).

7. Minor presentation items: consistently pair the "59%" relative figure with the absolute values (0.181 to 0.288); ensure the UMAP figures (Figures 7 and 8) are legible and cross-referenced to the quantitative separability metrics in Table 3; and consider moving key negative-result summaries (Figure 4) earlier to reinforce the honest scoping.

---

### Review · Reviewer_s4nm · 2026-07-06

**Summary Of Contributions:**

This manuscript introduces a lightweight post-training pipeline to inject knowledge of protein 3D structures into pre-trained protein language models. During the inference stage, there is no need for structural inputs. The major novelty includes a dual-task framework combining cross-protein latent contrastive alignment with intra-protein structural token prediction. A residue loss selection module is used to mitigate low-quality PDB structural noise. The proposed method is validated across multiple tasks.

**Audience:**

Yes

**Audience Explanation:**

it fits the scope

**Claims And Evidence:**

Yes

**Claims Explanation:**

this work contains critical conceptual, experimental, benchmarking, mechanistic, and presentation weaknesses that should be fully resolved

**Requested Changes:**

This manuscript introduces a lightweight post-training pipeline to inject knowledge of protein 3D structures into pre-trained protein language models. During the inference stage, there is no need for structural inputs. The major novelty includes a dual-task framework combining cross-protein latent contrastive alignment with intra-protein structural token prediction. A residue loss selection module is used to mitigate low-quality PDB structural noise. The proposed method is validated across multiple tasks. Still, this work contains critical conceptual, experimental, benchmarking, mechanistic, and presentation weaknesses that should be fully resolved.

Major:

1.	The proposed latent-level contrastive loss only adds bidirectional a2g/g2a matching but does not propose a fundamentally new contrastive paradigm. Residue-level cross-modal alignment itself is not a unique contribution. The physical-level task also lacks novel geometric tokenization or geometric-equivariant constraints.

2.	Structure-aware training with sequence-only inference is a major advantage, which has been partially demonstrated in recent works. It is suggested to clearly explain how dual-task combination delivers benefits unavailable from single-task baselines beyond minor ablation gains.

3.	The authors show a consistent rise in PPL after structure alignment. But there is no mechanistic explanation for this tradeoff. Controlled distribution shift ablation experiments to disentangle confounding factors is recommended.

4.	There lacks quantitative causal analysis linking latent embedding alignment to downstream performance gains.

5.	Why does structural alignment fail to boost functional prediction?

6.	The LLPS condensate classification benchmark only uses predicted ESMFold structures for competing structure-conditioned models as a comparison control. It is suggested to include experiments conducted with experimentally resolved disordered protein conformations or conformational ensembles.

7.	Experiments on larger models with 1B+ parameters are suggested to validate whether structure alignment gains persist at frontier pLM scales.

8.	The pGNN ablation omits other state-of-the-art structural encoders.

Minor:

1.	Bootstrap confidence intervals are provided for some tasks but omitted for many functional benchmarks.

2.	P-values for performance differences between base and aligned models are not systematically reported.

3.	The data and code availability statement only promises release post-publication, with no temporary access, supplementary algorithm pseudocode, or open-source demo notebooks provided for peer review reproducibility checks.

---

### Review · Reviewer_asHt · 2026-07-16

**Summary Of Contributions:**

This paper proposes a post-training framework for incorporating structural knowledge into sequence-only protein language models. It makes three main contributions:

- Dual-task structure alignment. The method combines residue-level contrastive alignment between protein language model representations and embeddings from a frozen protein graph neural network with a structure-token prediction objective. The original masked-language-modeling objective is retained, allowing the resulting models to operate using sequence alone at inference time.

- Residue-level loss selection. To reduce the influence of unreliable structures in the PDB, the authors train a smaller reference model on a high-quality structural subset and use excess loss to select residues that are intended to be both reliable and challenging for the current model.

- Empirical evaluation. The method is applied to ESM2 and AMPLIFY across multiple model sizes. The experiments report improvements primarily on structure-related tasks and aggregate zero-shot ProteinGym mutation-effect prediction, while results on broader functional-property benchmarks are mostly limited or inconclusive and pseudo-perplexity increases after alignment.

Key strengths:

- The sequence-only inference setting is practically useful, particularly when experimental or predicted structures are unavailable or unreliable.

- The evaluation covers two protein language model families, several model sizes, a broad collection of downstream tasks, and multiple ablation studies.

- The improvements on contact prediction, secondary-structure prediction, supervised stability prediction, and aggregate ProteinGym performance are promising, and the paper reports mixed or negative results rather than claiming uniform gains.

Key weaknesses:

- The implementation and computational feasibility of the dense residue-level contrastive objective are insufficiently explained given the reported batch size and sequence length.

- Potential overlap between the PDB alignment corpus and downstream evaluation proteins is not adequately audited, and several contact-prediction and stability metrics are inconsistently labeled across tables and figures.

- Some statistical analyses do not account for paired observations or correlations among CASP16 chains from the same complex. The residue-loss-selection mechanism also lacks direct validation against local structural quality, while the claims regarding efficiency, preservation of language-modeling capability, and increased biophysical understanding are stronger than the current evidence supports.

**Audience:**

Yes

**Audience Explanation:**

The paper studies how to incorporate structural knowledge into protein language models while retaining sequence-only inference. The proposed method and its results on structural and mutation-effect tasks should interest researchers in protein representation learning and computational biology.

**Broader Impact Concerns:**

Nope

**Claims And Evidence:**

No

**Claims Explanation:**

The paper provides encouraging evidence that the proposed structure-alignment method improves several structure-related tasks, and the experimental evaluation is broad, covering two model families, multiple model sizes, downstream benchmarks, and ablation studies. However, I believe a few additional clarifications and focused analyses would make the main conclusions more convincing.

1. It would be helpful to report the sequence or homology overlap between the PDB alignment corpus and the downstream evaluation sets. The date-based split for CASP16 is a useful precaution, but an explicit overlap analysis would further confirm that the improvements are not influenced by closely related proteins in the alignment data.

2. The existing ablations support the contribution of the latent-level and physical-level objectives. One additional control, such as shuffled sequence--structure pairings or randomized structure tokens, would help verify that the gains come from meaningful structural correspondence rather than from additional post-training or auxiliary-task regularization.

3. The CASP16 result is promising, but the evaluation contains 51 chains from only 13 complexes. Reporting confidence intervals at the complex or target level, in addition to the current chain-level analysis, would provide a more conservative estimate of uncertainty.

4. Some evaluation metrics appear to be labeled differently across tables and figures, particularly P@L versus P@L/5 for contact prediction and accuracy versus Spearman correlation for Stability. Clarifying these definitions would make the reported improvements easier to interpret.

5. Some conclusions could be stated more cautiously. In particular, pseudo-perplexity increases after alignment, and many broader functional-property results are statistically inconclusive. The results therefore support improved performance on selected structure-related and mutation-effect tasks, while the claims about fully preserving language-modeling ability or broadly increasing biophysical understanding may require further evidence.

**Requested Changes:**

The following changes would improve the clarity and reliability of the paper.

1. Please report the sequence or homology overlap between the PDB alignment corpus and the downstream evaluation sets. The date-based split for CASP16 is helpful, but an explicit overlap analysis would provide stronger evidence that the improvements are not influenced by closely related proteins in the alignment data.

2. Please clarify and correct the evaluation metrics used throughout the paper. In particular, contact prediction is reported as P@L in some places and P@L/5 in others, while Stability is described using both accuracy and Spearman correlation. The differences between the contact results reported in the main table and the ablation table should also be explained.

3. Please strengthen the statistical analysis of the main comparisons. Since the aligned and baseline models are evaluated on the same examples, paired confidence intervals or paired tests would be preferable. For CASP16, reporting uncertainty at the complex or target level, rather than only at the chain level, would better account for correlations among chains from the same complex.


4. Please add one additional control to verify that the gains arise from meaningful structural alignment, such as shuffled sequence--structure pairings or randomized structure tokens. For the residue-loss-selection module, a random-selection baseline using the same selection ratio would also help establish the benefit of the proposed strategy.

5. Please provide additional implementation details for the residue-level contrastive objective, including the actual contrastive microbatch size and how the potentially large residue similarity matrix is computed. It would also be helpful to clarify the computational overhead of the reference model and to moderate claims about fully preserving language-modeling capability, given the reported increase in pseudo-perplexity.

Overall, the experimental evaluation is broad and the results are promising. These focused clarifications and analyses would make the main conclusions substantially more convincing without requiring a major expansion of the benchmark suite.